# Stochastic adaptive Gradient Descent without descent

## Abstract

We introduce a new adaptive step-size strategy for convex optimization with stochastic gradient that exploits the local geometry of the objective function only by means of a first-order stochastic oracle and without any hyper-parameter tuning. The method comes from a theoretically-grounded adaptation of the Adaptive Gradient Descent Without Descent method to the stochastic setting. We prove the convergence of stochastic gradient descent with our step-size under various assumptions, and we show that it empirically competes against tuned baselines.

## 1 Introduction

We consider the stochastic convex optimization problem

$$\min_{x \in \mathbb{R}^d} f(x) = \min_{x \in \mathbb{R}^d} \mathbb{E}\left[f_\xi(x)\right], \tag{1}$$

where the expectation is taken with respect to some random variable $\xi$ such that, $f$ and $f_\xi : \mathbb{R}^d \to \mathbb{R}$ are convex, differentiable and lower-bounded functions for all $\xi$. We furthermore assume that $f$ has at least one minimizer $x^\star \in \mathbb{R}^d$, and we denote its smallest value by $f^\star = f(x^\star)$. When $\xi$ follows a uniform distribution on subsets of $\{1, \ldots, N\}$ (for some integer $N > 0$), (1) reads

$$\min_{x \in \mathbb{R}^d} f(x) = \min_{x \in \mathbb{R}^d} \frac{1}{N} \sum_{\ell=1}^{N} f_\ell(x), \tag{2}$$

where $f_1, \ldots, f_N$ is a set of convex, differentiable and lower-bounded functions.

### 1.1 Main contributions

We introduce a new adaptive step-size strategy for stochastic gradient descent (SGD) to tackle (1). We build upon the adaptive strategy introduced by Malitsky & Mishchenko (2020) for deterministic (full batch) convex optimization. They propose a step-size that automatically adapts to the local geometry of $f$ only by means of first-order oracle calls (*i.e.,*gradient evaluations). Their method has the notable advantages of not requiring any hyper-parameter tuning nor the global Lipschitz smoothness of the gradient of the objective function (Definition 1). While a stochastic version of the algorithm has been suggested in (Malitsky & Mishchenko, 2020, Section 3), it does not inherit the main advantages from the deterministic version: its implementation requires the careful tuning of a hyper-parameter, crucial to guarantee convergence. In contrast, we propose a different stochastic adaptation of their method that preserves the aforementioned advantages in the deterministic setting. As illustrated on Figure 1, our algorithm converges without requiring the tuning of any hyper-parameter, under various standard assumptions and models for stochastic optimization.

### 1.2 Stochastic Adaptive Descent

We consider a sequence $(\xi_k)_{k \in \mathbb{N}}$ of independent and identically distributed copies of $\xi$. For any random variable $Z$, $\mathbb{E}_{k-1}[Z]$ denotes the conditional expectation $\mathbb{E}[Z \mid \mathcal{F}_{k-1}]$, where $\mathcal{F}_{k-1}$ is the filtration up to iteration $k-1$. In particular, for all $k \in \mathbb{N}$, we assume access to a stochastic gradient oracle $\nabla f_{\xi_k}$ such that $\forall x \in \mathbb{R}^d$, $\mathbb{E}_{k-1}[\nabla f_{\xi_k}(x)] = \nabla f(x)$. We fix an initialization $x_0 \in \mathbb{R}^d$ and consider the stochastic recursion

$$x_{k+1} = x_k - \lambda_k \nabla f_{\xi_k}(x_k), \tag{3}$$

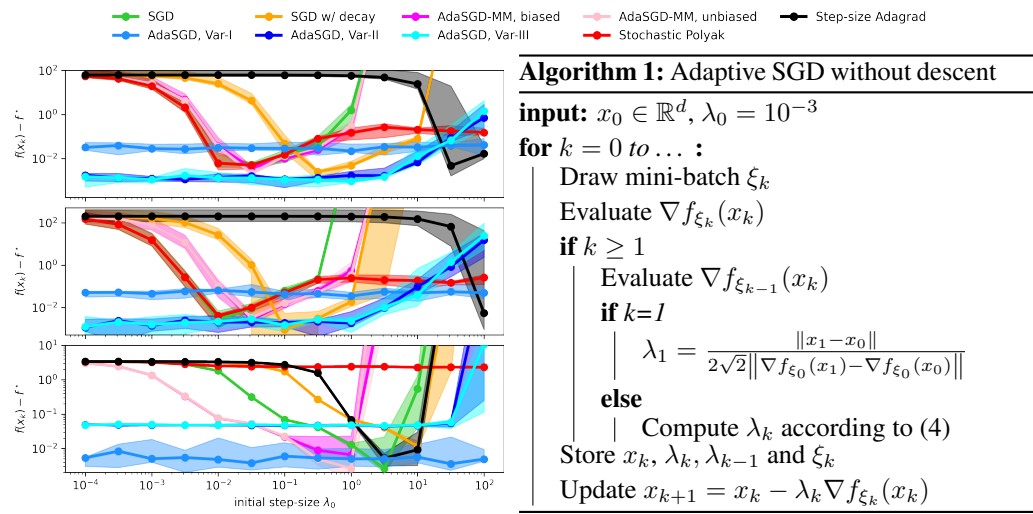

**Algorithm 1:** Adaptive SGD without descent

**input:** $x_0 \in \mathbb{R}^d$, $\lambda_0 = 10^{-3}$

**for** $k = 0$ *to* $\ldots$ **:**
    Draw mini-batch $\xi_k$
    Evaluate $\nabla f_{\xi_k}(x_k)$
    **if** $k \geq 1$
        Evaluate $\nabla f_{\xi_{k-1}}(x_k)$
        **if** *k=1*
            $\lambda_1 = \frac{\|x_1 - x_0\|}{2\sqrt{2}\|\nabla f_{\xi_0}(x_1) - \nabla f_{\xi_0}(x_0)\|}$
        **else**
            Compute $\lambda_k$ according to (4)
    Store $x_k, \lambda_k, \lambda_{k-1}$ and $\xi_k$
    Update $x_{k+1} = x_k - \lambda_k \nabla f_{\xi_k}(x_k)$

Figure 1: Influence of the initial step-size $\lambda_0$ on the optimality gap $f(x_k) - f^\star$ after 100 epochs. Each figure represents a different problem (linear, ridge and Poisson regression, see details in Section 4). Each experiment was ran five times, solid curve: median value , area: gap between best and worst values reached. Our methods (AdaSGD, the blue curves) always perform well with small values of $\lambda_0$ and thus do not require step-size tuning to compete with optimally-tuned SGD and "AdaSGD-MM" from Malitsky & Mishchenko (2020).

where step-size $\lambda_k > 0$ is a (possibly random) variable that is assumed to be independent of $\xi_k$ conditionally on $\mathcal{F}_{k-1}$. The above notation has been chosen to be consistent with those in Malitsky & Mishchenko (2020) to ease the comparison.

Our main contribution is to prove the benefits of the following adaptive step-sizes $\lambda_k$:

$$\hat{L}_{k-1} = \frac{\left\|\nabla f_{\xi_{k-1}}(x_k) - \nabla f_{\xi_{k-1}}(x_{k-1})\right\|}{\|x_k - x_{k-1}\|},$$

$$\lambda_k = \begin{cases} \min\left(\frac{1}{2\sqrt{2}\hat{L}_{k-1}} \quad , \quad \lambda_{k-1}\sqrt{1 + \frac{\lambda_{k-1}}{\lambda_{k-2}}}\right) & \text{(Variant \textbf{I})} \\ \min\left(\frac{1}{2\sqrt{2}\hat{L}_{k-1}}\frac{1}{k^{1/2+\delta}} \quad , \quad \lambda_{k-1}\sqrt{1 + \frac{\lambda_{k-1}}{\lambda_{k-2}}}\right) & \text{(Variant \textbf{II})} \\ \min\left(\frac{1}{2\sqrt{2}\hat{L}_{k-1}}\frac{1}{k^{1/2+\delta}} \quad , \quad \lambda_{k-1}\sqrt{1 + \left(1 - \frac{1}{k^{1/2+\delta}}\right)\frac{\lambda_{k-1}}{\lambda_{k-2}}}\right) & \text{(Variant \textbf{III})} \end{cases} \quad (4)$$

for all $k \geq 2$, with $\lambda_0 > 0$ and $0 < \delta < 1/2$. The choice (4) adapts locally to the geometry of the functions $f_\xi$. We propose three variants of our method that have different convergence properties, and recommend using Variant-**III** which has the strongest theoretical guarantees. The parameter $\delta > 0$ is not to be considered as a real hyper-parameter: it is needed for the theoretical convergence analysis, but can be taken *arbitrarily small*.

**Remark 1.1.** *Algorithm 1 is not exactly "parameter-free" as it features an initial $\lambda_0$. Yet, we argue it is "tuning-free" since taking a very small value of $\lambda_0$ is always as least as good as large ones, as highlighted in Figure 1 and also noted by Malitsky & Mishchenko (2020) in the deterministic setting.*

When the $f_\xi$ are $\mu$-strongly convex with $L$-Lipschitz gradient, we prove that variant-**III** of (4) provides (3) with the convergence rate $\mathbb{E}\left[\|x_{k+1} - x^\star\|^2\right] \leq \frac{C}{k^{1/2+\delta}}$, for all $k \geq k_0$, with $C > 0$ and $k_0 \geq 3$, *without any knowledge of the constants $\mu$ and $L$.* We provide additional guarantees beyond the globally Lipschitz continuous and strongly convex settings later in Section 3. We stress that (4) does not involve any scaling parameter to tune based on the curvature of the $f_\xi$'s which most choices for SGD typically require, including the approach in Malitsky & Mishchenko (2020) in the stochastic case. This is possible at the price of one extra stochastic gradient evaluation at each iteration. The same drawback exists with the stochastic approach of Malitsky & Mishchenko (2020), and it is

inherent to the idea they propose. However, this additional cost has to be balanced with the cost of hyper-parameter tuning (Sivaprasad et al., 2020).

### 1.3 RELATED WORK

Tuning the step-size in gradient descent (GD) has been the topic of a large body of literature. Without being exhaustive, we review the related work on adaptive step-sizes, with a focus on SGD.

**Further adaptation of Malitsky & Mishchenko (2020)** In the deterministic setting, Malitsky & Mishchenko (2020) introduced an adaptive version of GD (here-after called AdaGD) that they later refined in (Malitsky & Mishchenko, 2024) to improve constant factors in the choice of the step-size. A proximal version has been developed (Latafat et al., 2024). As far as we know, there has been little work on adapting AdaGD to the stochastic setting (beyond the heuristics in Malitsky & Mishchenko 2020). One exception is the work of Defazio et al. (2022), which takes inspiration from AdaGD, but which derives a rather different algorithm, that only allows for increasing step-sizes. In contrast, our method adapts to the local geometry at every iteration which generates non-monotone step-sizes.

**Adaptive methods.** Adaptive methods are a very active topic due to their extensive use for training neural networks (Duchi et al., 2011; McMahan & Streeter, 2010; Kingma & Ba, 2015; Tieleman & Hinton, 2012). However, they still feature a step-size parameter (and possibly others), whose choice significantly affects the performances (Sivaprasad et al., 2020). Beyond AdaGD, many other strategies exist in the deterministic setting (Li & Lan, 2023; Lan et al., 2023; Khaled et al., 2023). In particular, AdaGD can be seen as a more stable version of the Barzilai-Borwein step-sizes (Barzilai & Borwein, 1988; Raydan, 1997). Stochastic adaptations of these step-sizes have been proposed for convex (Tan et al., 2016) and non-convex optimization (Robles-Kelly & Nazari, 2019; Castera et al., 2022), but they always mitigate the instability of the methods through additional hyper-parameters. Overall, we stress that the stochastic setting possesses additional difficulties that make it significantly more challenging (Orabona & Cutkosky, 2020).

**Parameter free algorithms and normalized gradients** Regarding the stochastic setting, the term *parameter-free* may have several meanings. It is often connected to online learning where one seeks adaptive strategies of methods that *provably* minimize $f$ as well as the optimally-tuned instance of the method (up to a factor), see *e.g.*, (Orabona, 2014; Khaled & Jin, 2024). The definition of parameter free may vary depending on the assumptions on the function, the geometric constants of the problems that are assumed to be known, and the oracles used (e.g., accessing $f_\xi$ or not). Despite the name, parameter-free algorithms may still assume additional knowledge of the problem, such as a bound on gradient norms, see *e.g.,* (Orabona, 2023, Remark 1). A significant part of the work on parameter-free method uses normalized gradient strategies, *à la* Adagrad (McMahan & Streeter, 2010; Duchi et al., 2011). They normalize step-sizes using past (stochastic) evaluations of gradients, which has a provable benefit (Li & Orabona, 2019). Without being exhaustive, existing approaches beyond the ones above include (Levy, 2017; Orabona & Pál, 2021; Faw et al., 2022; Ivgi et al., 2023). We refer to Orabona (2023) for a detailed discussion on gradient normalization. When an upper-bound on gradient norms is known, the coin-betting strategy (Orabona & Pál, 2016; Orabona & Tommasi, 2017) provides acceleration by estimating the initial optimality gap $\|x_0 - x^\star\|$, see (Orabona, 2023) and references therein for more details on this line of work. In this setting, a close-to-optimal bisection method has been developed (Carmon & Hinder, 2022). When designing algorithms that can additionally access function values, the family of Polyak step-sizes (Polyak, 1987) can be considered, even though it was originally designed for deterministic optimization. This kind of methods has recently been adapted to the stochastic setting (Hazan & Kakade, 2019; Loizou et al., 2021), but it requires knowledge of the optimal values of the functions $(f_\xi)_\xi$. A different approach with access to function values has been proposed in (Chen et al., 2019). If the functions $f$ and the $f_\xi$ are themselves Lipschitz continuous, recent works show that this can be leveraged to design adaptive strategies in deterministic and stochastic settings (Cutkosky, 2019; Defazio & Mishchenko, 2023; Mishchenko & Defazio, 2024).

**Difference with parameter-free approaches** Although our work is connected to parameter-free considerations, we do not focus on theoretical comparisons with optimally tuned SGD. As a matter of fact, even in the deterministic setting, no theoretical acceleration was proved by Malitsky & Mishchenko (2020) compared to GD. Instead, our work focuses on adapting AdaGD to the stochastic setting while preserving the benefits of the method (see Section 1.1).

**Other related approaches** Line-search is a usual way to alleviate step-size selection. It is however difficult to adapt to the stochastic setting. Existing approaches (Byrd et al., 2012; Franchini et al., 2023) often require auxiliary hyper-parameters for the sake of stability (similarly to the aforementioned Barzilai-Borwein methods). Beyond GD, inertial methods are ubiquitous in optimization as they allow achieving optimal rates for convex (Nesterov, 1983) and strongly convex optimization (Nesterov, 2013, Algorithm 2.2.11). Adaptivity can also help to select the additional parameters these methods feature (Barré et al., 2020; Maier et al., 2023) or be used for adaptive restart strategies (Aujol et al., 2024; 2025). Finally, *universal methods* (Nesterov, 2015; Li & Lan, 2023; Grimmer, 2024) are worth mentioning. They relax the Lipschitz continuity of the gradient by assuming it has an (unknown) level of Hölder continuity. They try to automatically adapt to this level of Hölder continuity, without knowing it. The focus is thus different than that of the parameter-free setting previously discussed.

**Convergence of SGD.** Despite the different theoretical focus, we do provide theoretical guarantees via the Robbins-Siegmund theorem (Robbins & Siegmund, 1971). The latter is among the main tools to prove convergence in the stochastic setting. Many other proof strategies are possible depending on the variance assumption of the stochastic algorithm considered, see the discussion in Cortild et al. (2025) for further details, and Garrigos & Gower (2023) for a recollection of proofs.

## 1.4 ORGANISATION OF THE PAPER

In Section 2, we propose the derivation of a Lyapunov function that is inspired by the one proposed in Malitsky & Mishchenko (2020) in a deterministic setting and that we adapt to a stochastic setting. Using such a Lyapunov function, we discuss in Section 3 various assumptions of the functions $f_\xi$ that allow to derive the convergence of the stochastic sequence $(x_k)_{k \in \mathbb{N}}$ using the Lipschitz-adaptive choice (4) for the step-size. Numerical experiments are reported in Section 4 to illustrate the performances of our approach over a range of stochastic optimization problems. Finally, some auxiliary results are postponed to a technical Appendix at the end of the paper.

## 2 DESIGN OF THE METHOD

The idea from Malitsky & Mishchenko (2020) comes from the derivation of an original Lyapunov function. We present the main idea of this derivation and where our approach departs from theirs. Although Lipschitz continuity is not directly used below, we recall it in Definition 1 in the Appendix.

### 2.1 PREVIOUS RESULTS

In the deterministic setting, Malitsky & Mishchenko (2020) propose to use GD (*i.e.,*(3) with full batch) algorithm to solve (2) with step-size

$$\lambda_k = \min\left\{ \frac{\|x_k - x_{k-1}\|}{2\|\nabla f(x_k) - \nabla f(x_{k-1})\|}, \lambda_{k-1}\sqrt{1 + \theta_{k-1}} \right\}, \text{ with } \theta_{k-1} = \frac{\lambda_{k-1}}{\lambda_{k-2}}. \tag{5}$$

To ease the reading, we use the same notation $(x_k)_{k \in \mathbb{N}}$ to denote the iterates both in stochastic and deterministic settings. The main idea is that for the step-size (5), the sequence defined for $k \in \mathbb{N}$ by

$$T_k = \|x_{k+1} - x^\star\|^2 + 2\left(\lambda_k\left(1 + \theta_k\right)\right)(f(x_k) - f^\star) + \frac{\|x_{k+1} - x_k\|^2}{2} \tag{6}$$

is non-increasing along the iterates of GD. We thus call $(T_k)_{k \in \mathbb{N}}$ a Lyapunov sequence (or function). This is key as the choice (5), does not guarantee the decay of $(f(x_k))_{k \in \mathbb{N}}$, which is the usual Lyapunov sequence used for GD. This motivated the name "adaptive gradient descent without descent" (AdaGD). Not ensuring descent at each iteration allows for a choice $\lambda_k$ that adapts to the local Lipschitz constant of $\nabla f$ as argued in Malitsky & Mishchenko (2020).

In the case of (3), the authors consider the choice $\lambda_k = \min\left\{\alpha\Lambda_k, \lambda_{k-1}\sqrt{1 + \theta_{k-1}}\right\}$ with either $\Lambda_k = \frac{\|x_k - x_{k-1}\|}{\|\nabla f_{\xi_k}(x_k) - \nabla f_{\xi_k}(x_{k-1})\|}$ or $\Lambda_k = \frac{\|x_k - x_{k-1}\|}{\|\nabla f_{\zeta_k}(x_k) - \nabla f_{\zeta_k}(x_{k-1})\|}$ with $\zeta_k$ an independent copy of $\xi_k$, and where $\alpha > 0$ is a hyper-parameter to be tuned. They call these choices "biased" and "unbiased" for a reason we explain later. In contrast, in (4), we evaluate the difference of gradients

on the *previous* random variable $\xi_{k-1}$: $\left\| \nabla f_{\xi_{k-1}}(x_k) - \nabla f_{\xi_{k-1}}(x_{k-1}) \right\|$. This is a small but crucial difference that allows getting rid of the hyper-parameter $\alpha$ their approach features, and whose choice depends on the (a priori unknown) smoothness constants of the functions $\nabla f_{\xi_k}$.

## 2.2 ADAPTIVE STOCHASTIC GRADIENT DESCENT

We derive a Lyapunov analysis similar to that in Malitsky & Mishchenko (2020), and highlight the main differences between our approach and the original one. In what follows, we assume *and ensure* that the step-size satisfies the following.

**Assumption 1.** *The step-size $\lambda_k > 0$ is independent of $\xi_k$ conditionally on $\mathcal{F}_{k-1}$.*

**Remark 2.1.** *At this stage Assumption 1 is required to derive the computations below, but we stress that we designed our step-sizes (4) so that it always holds true.*

Let $k \geq 2$, as in the deterministic case we begin with the decomposition

$$\|x_{k+1} - x^\star\|^2 = \|x_{k+1} - x_k\|^2 + 2\langle x_{k+1} - x_k, x_k - x^\star\rangle \quad + \|x_k - x^\star\|^2$$
$$= \|x_{k+1} - x_k\|^2 - 2\langle \lambda_k \nabla f_{\xi_k}(x_k), x_k - x^\star\rangle + \|x_k - x^\star\|^2.$$

We introduce the notation $\Delta_k = x_{k+1} - x_k$. Taking conditional expectations on both sides of the above equality, we obtain that

$$\mathbb{E}_{k-1}\left[\|x_{k+1} - x^\star\|^2\right] = \mathbb{E}_{k-1}\left[\|\Delta_k\|^2\right] - 2\langle \mathbb{E}_{k-1}\left[\lambda_k \nabla f_{\xi_k}(x_k)\right], x_k - x^\star\rangle + \|x_k - x^\star\|^2. \quad (7)$$

Focusing on the second term in the right-hand side: due to Assumption 1 the conditional independence between $\xi_k$ and $\lambda_k$ allows splitting the conditional expectation into a product. This is the reason for the terminology "unbiased" as it then allows to use the property that $\mathbb{E}_{k-1}\left[\nabla f_{\xi_k}(x_k)\right] = \nabla f(x_k)$, meaning that $\nabla f_{\xi_k}(x_k)$ is an unbiased estimator of $\nabla f(x_k)$. We obtain

$$\langle \mathbb{E}_{k-1}\left[\lambda_k \nabla f_{\xi_k}(x_k)\right], x_k - x^\star\rangle = \mathbb{E}_{k-1}\left[\lambda_k\right]\langle \nabla f(x_k), x_k - x^\star\rangle \geq \mathbb{E}_{k-1}\left[\lambda_k\right](f(x_k) - f^\star), \quad (8)$$

where we used the convexity of $f$ in the last inequality. The function values $f(x_k)$ appears above, so we expect the quantity $f(x_{k-1})$ to appear as well later in our Lyapunov analysis.

Now focusing on $\Delta_k$, it is trivial but convenient to write $\|\Delta_k\|^2 = 2\|\Delta_k\|^2 - \|\Delta_k\|^2$ and use the identity $2\|\Delta_k\|^2 = -2\langle \lambda_k \nabla f_{\xi_k}(x_k), \Delta_k\rangle$. We then make an extra stochastic gradient appear,

$$-2\langle \lambda_k \nabla f_{\xi_k}(x_k), \Delta_k\rangle = \underbrace{-2\lambda_k \langle \nabla f_{\xi_k}(x_k) - \nabla f_{\xi_{k-1}}(x_{k-1}), \Delta_k\rangle}_{:=A_k} \underbrace{-2\lambda_k \langle \nabla f_{\xi_{k-1}}(x_{k-1}), \Delta_k\rangle}_{:=B_k}$$
$$(9)$$

This is a first pivoting choice in our analysis, it yields two terms that we analyze separately.

### 2.2.1 BOUNDING $B_k$

We first use (3) to substitute $\nabla f_{\xi_{k-1}}(x_{k-1})$ in $B_k$:

$$B_k = -2\lambda_k \left\langle -\frac{1}{\lambda_{k-1}}(x_k - x_{k-1}), -\lambda_k \nabla f_{\xi_k}(x_k)\right\rangle = -2\frac{\lambda_k^2}{\lambda_{k-1}}\langle x_k - x_{k-1}, \nabla f_{\xi_k}(x_k)\rangle$$

We take conditional expectation and use again Assumption 1 and the convexity of $f$ to get

$$\mathbb{E}_{k-1}\left[B_k\right] = -2\frac{\mathbb{E}_{k-1}\left[\lambda_k^2\right]}{\lambda_{k-1}}\langle x_k - x_{k-1}, \nabla f(x_k)\rangle \leq 2\frac{\mathbb{E}_{k-1}\left[\lambda_k^2\right]}{\lambda_{k-1}}(f(x_{k-1}) - f(x_k)),$$
$$= 2\frac{\mathbb{E}_{k-1}\left[\lambda_k^2\right]}{\lambda_{k-1}}(f(x_{k-1}) - f^\star) - 2\frac{\mathbb{E}_{k-1}\left[\lambda_k^2\right]}{\lambda_{k-1}}(f(x_k) - f^\star) \quad (10)$$

We obtain a term with $f(x_{k-1})$ that will be balanced by (8) and will result in a condition between $\lambda_k$ and $\lambda_{k-1}$, similar to that in (5). Plugging $\nabla f_{\xi_{k-1}}(x_{k-1})$ into (9) allows our analysis to remain faithful to the approach followed in the deterministic case.

### 2.2.2 Dealing with the term $A_k$

To handle $A_k$ in (9) we start by using $|\langle x, y \rangle| \leq \|x\|^2 + \frac{\|y\|^2}{4}, \forall x, y \in \mathbb{R}^d$:

$$-2\lambda_k \left\langle \nabla f_{\xi_k}(x_k) - \nabla f_{\xi_{k-1}}(x_{k-1}), \Delta_k \right\rangle \leq 2\lambda_k^2 \left\| \nabla f_{\xi_k}(x_k) - \nabla f_{\xi_{k-1}}(x_{k-1}) \right\|^2 + \frac{\|\Delta_k\|^2}{2}.$$

The second term in the right-hand side above will be compensated by the $- \|\Delta_k\|^2$ obtained above (9). Handling the first term is more involved. We plug again an extra stochastic gradient:

$$
\begin{aligned}
&\left\| \nabla f_{\xi_k}(x_k) - \nabla f_{\xi_{k-1}}(x_{k-1}) \right\|^2 \\
&= \left\| \nabla f_{\xi_k}(x_k) - \nabla f_{\xi_{k-1}}(x_k) + \nabla f_{\xi_{k-1}}(x_k) - \nabla f_{\xi_{k-1}}(x_{k-1}) \right\|^2 \\
&\leq 2\underbrace{\left\| \nabla f_{\xi_k}(x_k) - \nabla f_{\xi_{k-1}}(x_k) \right\|^2}_{\text{induced by sampling}} + 2\underbrace{\left\| \nabla f_{\xi_{k-1}}(x_k) - \nabla f_{\xi_{k-1}}(x_{k-1}) \right\|^2}_{\text{induced by curvature}},
\end{aligned}
\tag{11}
$$

where we used $\|x + y\|^2 \leq 2\|x\|^2 + 2\|y\|^2$. The decomposition (11) is a subtle but crucial difference with Malitsky & Mishchenko (2020) in the stochastic setting. As highlighted in (11) it allows bounding the difference of stochastic gradients by a sampling error and a "curvature-induced" term. In the deterministic setting ($\xi_k = \xi_{k-1}$) the first term vanishes and the curvature term boils down to the difference of gradients appearing in (5). This leads to our first condition on $\lambda_k$ in (4):

$$\lambda_k \leq \frac{\|x_k - x_{k-1}\|}{2\sqrt{2} \left\| \nabla f_{\xi_{k-1}}(x_k) - \nabla f_{\xi_{k-1}}(x_{k-1}) \right\|}
\tag{12}$$

that allows bounding the curvature term: $4\lambda_k^2 \left\| \nabla f_{\xi_{k-1}}(x_k) - \nabla f_{\xi_{k-1}}(x_{k-1}) \right\|^2 \leq \frac{\|\Delta_{k-1}\|^2}{2}$. Therefore, using again Assumption 1, we obtain

$$\mathbb{E}_{k-1}[A_k] \leq \mathbb{E}_{k-1}\left[\lambda_k^2\right] \mathbb{E}_{k-1}\left[\left\| \nabla f_{\xi_k}(x_k) - \nabla f_{\xi_{k-1}}(x_k) \right\|^2\right] + \frac{\|\Delta_{k-1}\|^2}{2} + \frac{\|\Delta_k\|^2}{2}.
\tag{13}$$

### 2.3 Decay of the Lyapunov function in Expectation

We use (10) and (13) in (9) to bound the first term in (7):

$$
\begin{aligned}
\mathbb{E}_{k-1}\left[\|\Delta_k\|^2\right] \leq &\mathbb{E}_{k-1}\left[\lambda_k^2\right] \mathbb{E}_{k-1}\left[\left\| \nabla f_{\xi_k}(x_k) - \nabla f_{\xi_{k-1}}(x_k) \right\|^2\right] + \frac{\|\Delta_{k-1}\|^2}{2} \\
&- \mathbb{E}_{k-1}\left[\frac{\|\Delta_k\|^2}{2}\right] + 2\frac{\mathbb{E}_{k-1}\left[\lambda_k^2\right]}{\lambda_{k-1}}(f(x_{k-1}) - f^\star) - 2\frac{\mathbb{E}_{k-1}\left[\lambda_k^2\right]}{\lambda_{k-1}}(f(x_k) - f^\star),
\end{aligned}
$$

which, going back to (7) and rearranging, leads to

$$
\begin{aligned}
\mathbb{E}_{k-1}&\left[\|x_{k+1} - x^\star\|^2\right] + 2\left(\mathbb{E}_{k-1}[\lambda_k] + \frac{\mathbb{E}_{k-1}\left[\lambda_k^2\right]}{\lambda_{k-1}}\right)(f(x_k) - f^\star) + \mathbb{E}_{k-1}\left[\frac{\|\Delta_k\|^2}{2}\right] \\
&\leq \|x_k - x^\star\|^2 + \frac{\|\Delta_{k-1}\|^2}{2} + 2\frac{\mathbb{E}_{k-1}\left[\lambda_k^2\right]}{\lambda_{k-1}}(f(x_{k-1}) - f(x_k)) \\
&\qquad\qquad\qquad + 4\mathbb{E}_{k-1}\left[\lambda_k^2\right] \mathbb{E}_{k-1}\left[\left\| \nabla f_{\xi_k}(x_k) - \nabla f_{\xi_{k-1}}(x_k) \right\|^2\right].
\end{aligned}
$$

We almost recovered an inequality on the sequence $(T_k)_{k \in \mathbb{N}}$ defined in (6). Our goal at this point is to make $(T_k)_{k \in \mathbb{N}}$ non-increasing, up to the last term above that we will handle later. This is where our second condition comes in. Assume that for all $k \geq 2$,

$$\mathbb{E}_{k-1}\left[\lambda_k^2\right] \leq \lambda_{k-1}^2 (1 + \theta_{k-1}) \quad \text{with} \quad \theta_{k-1} = \lambda_{k-1}/\lambda_{k-2},
\tag{14}$$

then, under this condition and (12), we just obtained our stochastic version of the Lyapunov descent from Malitsky & Mishchenko (2020).

**Proposition 2.2.** *Under Assumption 1 and conditions* (12) *and* (14)*, $T_k$ defined in* (6) *satisfies*

$$\mathbb{E}_{k-1}\left[T_k\right] \leq T_{k-1} + 4\mathbb{E}_{k-1}\left[\lambda_k^2\right]\mathbb{E}_{k-1}\left[\left\|\nabla f_{\xi_k}(x_k) - \nabla f_{\xi_{k-1}}(x_k)\right\|^2\right]. \qquad (15)$$

**Remark 2.3.** *Condition* (14) *involves an expectation, which is inconvenient in practice. This is easily fixed by taking $\lambda_k \leq \lambda_{k-1}\sqrt{1 + \theta_{k-1}}$, as we do in* (4)*, which obviously satisfies* (14)*.*

In particular for the three variants that we propose in (4), conditions (12) and (14) are fulfilled.

**Corollary 2.4.** *For any of the three choices in* (4)*, Proposition 2.2 holds true for Algorithm 1.*

To conclude this section, we importantly note that our analysis unifies the deterministic and stochastic settings. Indeed, when $\xi_k = \xi_{k-1}$, (15) simply reads $T_k \leq T_{k-1}$ and conditions (12) and (14) are exactly (5), up to a $\sqrt{2}$ factor. Therefore, the main difference in the stochastic setting is the term $\mathbb{E}_{k-1}\left[\lambda_k^2\right]\mathbb{E}_{k-1}\left[\left\|\nabla f_{\xi_k}(x_k) - \nabla f_{\xi_{k-1}}(x_k)\right\|^2\right]$ in (15), which can informally be interpreted as the variance of the noise induced by stochastic approximations. To derive convergence results one needs to control this term, a classical challenge in stochastic optimization. Next section is devoted to this.

## 3 CONVERGENCE ANALYSIS

Throughout this section, we consider Algorithm 1 with the three choices of step-size (4). Remark that with choice (4), $\lambda_k$ is not a random variable conditionally on $\mathcal{F}_{k-1}$, so we simply have $\mathbb{E}_{k-1}\left[\lambda_k^2\right] = \lambda_k^2$. Furthermore Assumption 1 holds true. We study the convergence of Algorithm 1, all the proofs of this Section are deferred to the appendix. According to Corollary 2.4, the sequence $(T_k)_{k\in\mathbb{N}}$ satisfies (15) for Algorithm 1 in the stochastic setting only by assuming *local* Lipschitz continuity of the $\nabla f_\xi$'s. However, the inequality (15) derived for $(T_k)_{k\in\mathbb{N}}$ features an additional term inherent to the stochastic case. The crux to obtain further convergence properties is to control this last term (15). First, we show in Lemma B.1 of Appendix B.1 that $\forall k \geq 1$,

$$\mathbb{E}_{k-1}\left[\left\|\nabla f_{\xi_k}(x_k) - \nabla f_{\xi_{k-1}}(x_k)\right\|^2\right]$$

$$\leq 4\mathbb{E}_{k-1}\left[\left\|\nabla f_{\xi_k}(x_k) - \nabla f_{\xi_k}(x^\star)\right\|^2\right] + 4\mathbb{E}_{k-1}\left[\left\|\nabla f_{\xi_{k-1}}(x_k) - \nabla f_{\xi_{k-1}}(x^\star)\right\|^2\right] + 4\sigma_k^2, \quad (16)$$

where $\sigma_k^2 = \mathbb{E}_{k-1}\left[\left\|\nabla f_{\xi_k}(x^\star)\right\|^2\right] + \mathbb{E}_{k-1}\left[\left\|\nabla f_{\xi_{k-1}}(x^\star)\right\|^2\right]$. The first two terms in (16) may vanish asymptotically if $x_k$ converged to $x^\star$, but $\sigma_k$ is bounded away from zero in general.[1] This evidences that ensuring convergence in the stochastic setting requires additional assumptions on the objective function $f$. We present three possible sets of assumptions that allow controlling (16).

**Assumption 2.** *One of the following assumptions below holds true.*

*(2-i) The function $f$ has the finite-sum structure* (2) *and each $\nabla f_\ell$ is $L_\ell$-Lipschitz continuous.*

*(2-ii) There exists $L > 0$ such that each $\nabla f_\xi$ is $L$-Lipschitz and $\sigma > 0$ such that $\forall k, \sigma_k^2 < \sigma^2$.*

*(2-iii) The function $f$ has the finite-sum structure* (2) *and the iterates $(x_k)_{k\in\mathbb{N}}$ are bounded.*

Any of these sets of properties provides a uniform control on the problematic term in (15). Assumption (2-ii) is necessary when one minimizes infinite sums, Assumption (2-i) requires global Lipschitz continuity for finite sums, which can be relaxed in (2-iii) but at the cost of assuming boundedness of the iterates, which can only be checked *a posteriori*.

**Lemma 3.1.** *Under the conditions in Assumption 2, $\forall k \geq 1$, there exists $L, \sigma > 0$ such that*

$$\lambda_k^2 \mathbb{E}_{k-1}\left[\left\|\nabla f_{\xi_k}(x_k) - \nabla f_{\xi_{k-1}}(x_k)\right\|^2\right] \leq 8\lambda_k^2 L^2 \left\|x_k - x^\star\right\|^2 + 4\lambda_k^2 \sigma^2$$

One can see that when $\lambda_k^2$ converges to zero "fast enough", we should obtain convergence of $(T_k)_{k\in\mathbb{N}}$. We now present the main convergence results of the paper in the next two sections.

---

[1] Except in the so-called interpolated case where $x^\star$ minimizes all the $f_\xi$'s.

## 3.1 Almost sure convergence

We begin with a general theorem.

**Theorem 3.2.** *Consider Algorithm 1 with step-size* (4). *If Assumption 2 holds true, then whenever the sequence of step-sizes* $(\lambda_k)_{k \in \mathbb{N}}$ *satisfies* $\sum_{k \geq 0} \lambda_k^2 < +\infty$ *almost surely, the Lyapunov sequence* $(T_k)_{k \in \mathbb{N}}$ *converges almost surely.*

The proof relies on the Robbins-Siegmund theorem (Robbins & Siegmund, 1971) that is recalled in the Appendix for completeness. Theorem 3.2 requires the sequence of step-sizes to be square-summable. This is true in the following settings (as proved in Appendix A), and possibly others:

**Case**-1 when there exists $\mu > 0$ such that each $f_\xi$ is $\mu$-strongly convex;

**Case**-2 in least-square regression: $f$ is a finite sum (2) and $\forall x \in \mathbb{R}^d, f_\xi(x) = (\langle w_\xi, x \rangle - y_\xi)^2$, for $w_\xi \in \mathbb{R}^d$ and $y_\xi \in \mathbb{R}$;

**Case**-3 more generally when $f$ is a finite sum (2) of *ridge* functions: $\forall x \in \mathbb{R}^d, f_\xi(x) = g_\xi(\langle w_\xi, x \rangle)$, and $g_\xi : \mathbb{R} \to \mathbb{R}$ is strongly convex.

We recall, that $f_\xi$ is $\mu$-strongly convex means $f_\xi - \frac{\mu}{2} \|\cdot\|^2$ is convex. **Case**-3 is a generalization of linear regression that has important applications in the literature. See Section A of the appendix for further discussion on these three cases. In these settings we obtain the convergence of the algorithm.

**Corollary 3.3.** *Consider Algorithm 1 with step-size variant-**II** or **III** from* (4). *Under Assumption 2, the sequence* $(T_k)_{k \in N}$ *defined in* (15) *converges almost surely for any of **Case**-1, **Case**-2 and **Case**-3.*

We just provided prominent practical cases in which Algorithm 1 converges without requiring any hyper-parameter tuning. We now show that it is actually possible to improve the results and derive rates of convergence in **Case**-1.

## 3.2 Convergence rates in expectation

When the functions $f'_\xi s$ are strongly convex (**Case**-1), Variant-**III** of step-size (4) provides the algorithm with a rate of convergence.

**Theorem 3.4.** *Consider **Case**-1 with Assumption 2. Then for Variant-**III** of the step-size* (4) *with any* $0 < \delta < 1/2$, *there exists* $C > 0$ *and* $k_0 \geq 3$, *such that the iterates of Algorithm 1 satisfy* $\mathbb{E}[T_k] \leq \frac{C}{k^{1/2+\delta}}, \quad \forall k \geq k_0$. *In particular the same rate holds for* $\mathbb{E}\left[\|x_{k+1} - x^\star\|^2\right]$.

We recommend using Variant-**III** of the step-size as it provides the strongest guarantees.

**Remark 3.5.** *The iteration* $k_0$ *after which this rate holds does not depend on* $\lambda_0$, *see Appendix B.2. This is not true for vanilla SGD with decreasing step-sizes, where one gets similar asymptotic convergence rates but only after some* $k_0$ *that depends on the initial step-size or that requires additional knowledge on the strong-convexity parameter (Moulines & Bach, 2011). Finding the right initial step-size for vanilla SGD, is also crucial to get good non-asymptotic performance in practice (Asi & Duchi, 2019).*

## 4 Numerical experiments

To assess the performance of AdaSGD, we consider six different stochastic convex differentiable optimization problems of the form (1), with synthetic and real data. We consider four types of loss functions that illustrate the possible different settings from Section 3: linear and ridge regression that fit **Case**-2 and **Case**-3, respectively; logistic regression that does not fit the cases considered, and Poisson regression where $\nabla f$ is locally Lipschitz continuous but not globally. We compare our method to SGD with fixes step-size $\lambda_0$ (no decay), SGD with decay: $\lambda_k = \frac{\lambda_0}{k^{1/2+\delta}}$ ($\delta = 10^{-2}$) and the two stochastic versions from Malitsky & Mishchenko (2020) (AdaSGD–MM) that also feature a tunable step-size $\lambda_0$ (previously denoted by $\alpha$ in Section 2.1). We also consider two additional methods with adaptive step-sizes: the step-size version of Adagrad and the stochastic Polyak step-size (Loizou et al., 2021). Note that these methods still feature a "step-size" like parameter (denoted

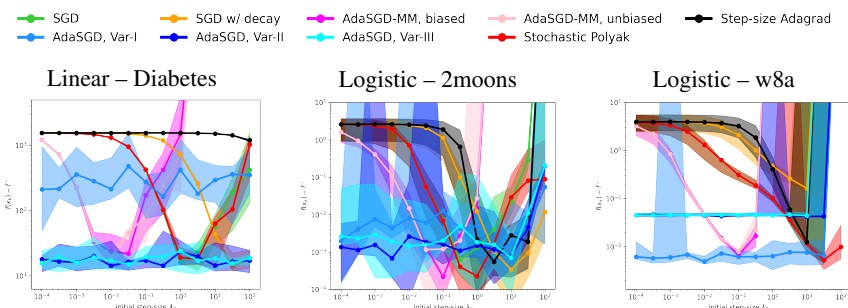

Figure 2: Experiments in the same setting as in Figure 1 on three other optimization problems. Except for a few outliers, out methods are significantly more stable to the choice of $\lambda_0$ than other algorithms and always work well for small values of $\lambda_0$.

by $\lambda_0$ in the figures for consistency). We provide further details on the experiments in Section D of the Appendix and a publicly available code.[2]

**Sensitivity to the initial step-size.** We first study how the choice of the initial step-size $\lambda_0$ affects each of the seven algorithms. The results for the first three problems on Figure 1 and the three others on Figure 2 show that AdaSGD is indeed much less sensitive to $\lambda_0$ than other algorithms (except for a few outliers in logistic regression, but the median remains stable even there). This significantly reduced sensitivity also does not seem to influence the performance as the value of $f$ achieved is comparable or better than that of optimally-tuned SGD. We also remark that despite being adaptive, the step-size Adagrad and Stochastic Polyak are more sensitivities to $\lambda_0$ than our method. We note nonetheless that Variant-**I** seems to have a significantly different behavior than Variants-**II** and **III** (which are relatively close). While Variant-**I** comes with less strong theoretical guarantees, it is sometimes better (or worse) than the other two variants depending on the problem. This is not surprising as vanilla SGD also peforms sometimes better with or without decay depending on the problem. Figures 1 and 2 clearly show that our method fixes the issue from the stochastic method of Malitsky & Mishchenko (2020) as for the latter, $\lambda_0$ has a crucial impact on performance. This is the main goal and contribution of this work. An important take-away for practitioners is that we see that *taking low values of $\lambda_0$ for AdaSGD seem to always be a good choice*, therefore we can use any variant of AdaSGD with (for example) $\lambda_0 = 10^{-3}$ and avoid tuning the step-size.

**Minimization performance.** We ran AdaSGD with the default value (*without tuning $\lambda_0$*) and compare it to the other algorithms that were tuned via a dense grid search on six problems. Results are shown on Figure 3. We note that variant-**I** sometimes struggles, which illustrates the importance of the decay introduced in Variants-**II** and **III**. They exhibit fewer oscillations due to the control of the noise (the additional term in (15)). Interestingly, Variant-**I** works better than the other two on Poisson regression, which is a problem where there is no global Lipschitz continuity of the gradients. We observe that the performance of AdaSGD is comparable to that of the other methods. The gain or loss is often marginal but AdaSGD has the significant benefit that no tuning is required to achieve this performance. AdaSGD requires two gradient evaluations per iteration but this is compensated by the absence of grid search, which itself requires many gradient oracle calls.

**Effect of the batch-size.** To investigate the effect that the size of the mini-batches have on the stability to $\lambda_0$, we repeated some of the experiments in the case of high noise (small mini-batches) and low noise (large mini-batches). The results presented in Figure 4 and Figure 5 in the appendix show that our conclusions remain the same: a small choice of $\lambda_0$ is always good for our three methods and they exhibit minimization performances comparable to those of most optimally-tuned methods considered.

---

[2]https://github.com/anonrepo555/AdaSGD_ICLR

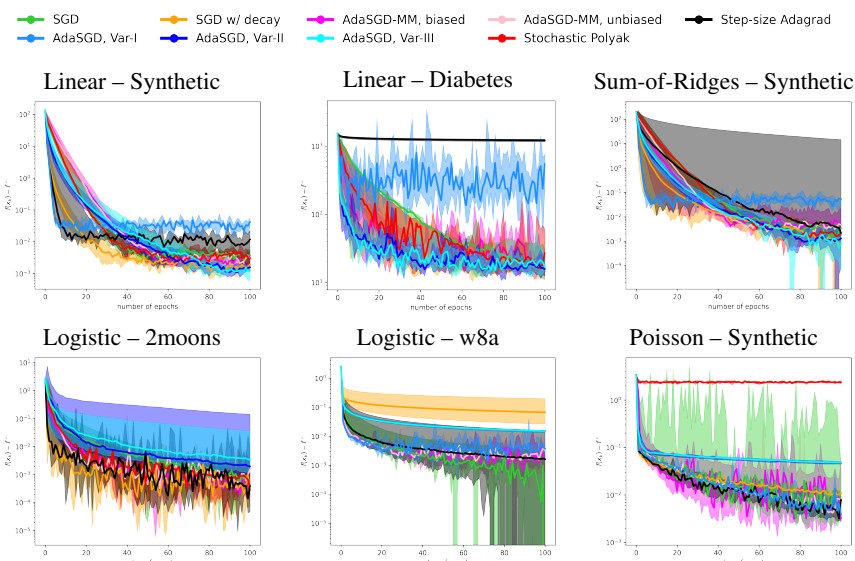

Figure 3: Evolution of the sub-optimality $f(x_k) - f^\star$ over the epochs. Comparison between the three variants of AdaSGD used *without tuning* and other methods that are tuned via a dense grid-search. Solid curves: median value across five runs, area: gap between best and worst values.

## 5 CONCLUSION

We introduced AdaSGD, an adaptation of AdaGD to the stochastic setting that coincides with AdaGD in the full-batch case. AdaSGD transfers the advantages of AdaGD to the stochastic setting, some of which were lost in the original stochastic adaptation of AdaGD. Our method does not rely on any heuristics and is rather derived via a careful Lyapunov analysis, resulting in three variants step-size strategies. We conducted a thorough analysis, showing convergence and rates for our methods. These guarantees however rely on stronger assumptions than in the deterministic setting, which is due to the noise induced by stochastic gradients. We show promising experiments that evidence that our primary goal is achieved: AdaSGD is very robust to the choice of the initial step-size without sacrificing minimization performance.

The main directions for further research revolve around relaxing the requirements related to convexity. In Appendix C, we provide partial theoretical results that suggest that the convergence results of Section 3 could possibly be extended to all convex functions. On the other-hand, adapting AdaSGD to non-convex functions would allow training neural networks but represents a significant challenge since the derivation of Section 2 relies heavily on the convexity inequality.

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

## A  ENSURING SQUARE-SUMMABILITY OF THE STEP-SIZES

Theorem 3.2 holds true when the sequence of step-sizes $(\lambda_k)_{k \in \mathbb{N}}$ is square-summable. We can guarantee this in **Case**-1, **Case**-2 and **Case**-3 as discussed in the main text. We now prove this and provide further discussion on these three settings.

### A.1  CASE-1: STRONG CONVEXITY

In **Case**-1, we assume that all the functions $f_\xi$ are strongly convex, which is a sub-class of convex functions. We recall that this means that there exists $\mu > 0$, such that for all $\xi$, and $x, y \in \mathbb{R}^d$,

$$f_\xi(y) \geq f_\xi(x) + \langle \nabla f_\xi(x), y - x \rangle + \frac{\mu}{2}\|x - y\|^2,$$

or equivalently

$$\langle \nabla f_\xi(x) - \nabla f_\xi(y), x - y \rangle \geq \mu\|x - y\|^2. \tag{17}$$

Note that we assume that all the functions $f_\xi$ have the same strong convexity parameter $\mu$. We could have equivalently assumed that there exists $\mu > 0$ such that each $f_\xi$ is $\mu_\xi$ strongly convex with $\mu_\xi \geq \mu > 0$. In the case where $f$ is a finite-sum of functions this is always possible (by taking $\mu = \min_\xi \mu_\xi$).

Strong convexity allows upper-bounding the step-sizes $(\lambda_k)_{k \in \mathbb{N}}$. Indeed, by using Cauchy-Schwarz inequality on (17), we get,

$$\|\nabla f_\xi(x) - \nabla f_\xi(y)\| \, \|x - y\| \geq \mu\|x - y\|^2.$$

So in particular, using this on the iterates $(x_k)_{k \in \mathbb{N}}$ and mini-batches $(\xi_k)_{k \in \mathbb{N}}$ in Algorithm 2, for all $k \geq 1$

$$\frac{\|x_k - x_{k-1}\|}{\left\|\nabla f_{\xi_{k-1}}(x_k) - \nabla f_{\xi_{k-1}}(x_{k-1})\right\|} \leq \frac{1}{\mu}. \tag{18}$$

Hence, from the definition of the step-size $\lambda_k$ in (4), we deduce that,

$$\lambda_k \leq \begin{cases} \frac{1}{2\sqrt{2}\mu}, & \text{(Variant \textbf{I})} \\ \frac{1}{k^{1/2+\delta}} \frac{1}{2\sqrt{2}\mu} & \text{(Variant \textbf{II} and \textbf{III})} \end{cases}$$

Therefore, for Variants **II** and **III** of the step-size, we can ensure that,

$$\sum_{k=0}^{+\infty} \lambda_k^2 \leq \sum_{k=0}^{+\infty} \frac{1}{k^{1+2\delta}} < +\infty.$$

Therefore, Theorem 3.2 holds true in **Case**-1. Why strong convexity may seem to be the natural assumption for ensuring square-summability, a similar result can be derived in non-strongly convex settings.

### A.2  CASE-2: LINEAR REGRESSION.

We now consider the classical machine learning problem of linear regression, where $f$ is a finite sum (2) and the $f_\ell$'s need not be strongly convex. We assume that for all $x \in \mathbb{R}^d$, the functions $(f_\ell)_{\ell \in \{1,\dots,N\}}$ take the following form:

$$f_\ell(x) = \frac{1}{2}(y_\ell - \langle w_\ell, x \rangle)^2,$$

where $w_\ell$ is a $d$-dimensional vector of predictors (with $w_\ell \neq 0$) and $y_\ell$ is a scalar response variable. The discussion can easily be extended to the case where $w_\ell$ are matrices and $y_\ell$ are vectors, which we do not consider for the sake of simplicity.

For all $\ell \in \{1, \dots, N\}$, and $x \in \mathbb{R}^d$, we have

$$\nabla f_\ell(x) = (\langle w_\ell, x \rangle - y_\ell)w_\ell, \tag{19}$$

and the Hessian of $f_\ell$ is given by

$$\nabla^2 f_\ell(x) = w_\ell w_\ell^T, \quad \text{for all } x \in \mathbb{R}^d.$$

Since $w_\ell w_\ell^T$ is a rank-1 matrix, the Hessian admits a zero eigenvalue (actually it even has $d-1$) so the function $f_\ell$ is not strongly convex.

We now use this on the iterates of Algorithm 1. For the sake of simplicity, we assume, without loss of generality, that the mini-batches are single valued, *i.e.* for all $k \in \mathbb{N}$, $\xi_k \in \{1, \dots, N\}$. Then remark that for any $k \in \mathbb{N}$ of Algorithm 1, (19) implies that

$$
\left\| \nabla f_{\xi_{k-1}}(x_k) - \nabla f_{\xi_{k-1}}(x_{k-1}) \right\|^2
$$
$$
= \left\| (y_{\xi_{k-1}} - \langle w_{\xi_{k-1}}, x_k \rangle) w_{\xi_{k-1}} - (y_{\xi_{k-1}} - \langle w_{\xi_{k-1}}, x_{k-1} \rangle) w_{\xi_{k-1}} \right\|^2
$$
$$
= \left\| \langle w_{\xi_{k-1}}, x_k - x_{k-1} \rangle w_{\xi_{k-1}} \right\|^2 = \langle w_{\xi_{k-1}}, x_k - x_{k-1} \rangle^2 \left\| w_{\xi_{k-1}} \right\|^2 .
$$

But remark that from the definition of Algorithm 1,

$$
x_k - x_{k-1} = -\lambda_{k-1} w_{\xi_{k-1}} = -\lambda_{k-1} \alpha_{k-1} w_{\xi_{k-1}},
$$

where $\alpha_{k-1} = (\langle w_{\xi_{k-1}}, x \rangle - y_{\xi_{k-1}})$. So inserting the above in (20),

$$
\left\| \nabla f_{\xi_{k-1}}(x_k) - \nabla f_{\xi_{k-1}}(x_{k-1}) \right\|^2 = \lambda_{k-1}^2 \alpha_{k-1}^2 \left\| w_{\xi_{k-1}} \right\|^6 . \tag{20}
$$

Then similarly,

$$
\left\| x_k - x_{k-1} \right\|^2 = \lambda_{k-1}^2 \alpha_{k-1}^2 \| w_{\xi_{k-1}} \|^2 ,
$$

So we obtain that

$$
\frac{\| x_k - x_{k-1} \|}{\left\| \nabla f_{\xi_{k-1}}(x_k) - \nabla f_{\xi_{k-1}}(x_{k-1}) \right\|} = \frac{1}{\| w_{\xi_{k-1}} \|^2} .
$$

Finally, by using the finite-sum structure, we denote by $\hat\mu = \max_{\ell \in \{1,\dots,N\}} \| w_\ell \|^2$ and we arrive to the same inequality as in (18), *without* relying on strong convexity. Therefore the same reasoning as in **Case**-1 applies here and we deduce that for Variants **II** and **III** of the step-size, $\sum_{k=0}^{+\infty} \lambda_k^2 < +\infty$.

### A.3 CASE-3: SUM OF RIDGE FUNCTIONS

As a third setting, we consider a generalization of linear regression where the function $f$ is a finite sum of convex ridge functions. More precisely, we assume that, for each $\ell \in \{1, \dots, N\}$, the function $f_\ell$ takes the form

$$
f_\ell(x) = g_\ell(\langle w_\ell, x \rangle),
$$

where $w_\ell$ is defined as in **Case**-2, and $g_\ell \colon \mathbb{R} \to \mathbb{R}$ is a known function that is differentiable and $\mu_\ell$-strongly convex on $\mathbb{R}$. This setting includes the previous example of linear regression with $g_\ell(u) = \frac{1}{2}(u - y_\ell)^2$, for all $u \in \mathbb{R}$.

Linear combinations of ridge functions constitute a functional class that is a central tool in the approximation theory of multivariate functions, and for the study of neural networks with a single or two hidden layers. For a survey on these topics we refer to (Ismailov, 2021). Using sums of ridge functions has also proved to be relevant for the regularization of inverse problems (Goujon et al., 2023), the study of convex optimization in the space of measures (Chizat, 2022), and for adversarial bandit problems (Rajaraman et al., 2024) to name a few.

Note that assuming that as for linear regression, the $f_\ell$ are not strongly convex (they are only in one direction given by $w_\ell$), and that $f$ needs not to be strongly convex as well. Hence, this setting remains different from **Case**-1.

Similarly to **Case**-2, for all $\ell \in \{1, \dots, N\}$ and $x \in \mathbb{R}^d$,

$$
\nabla f_\ell(x) = g_\ell'(\langle w_\ell, x \rangle) w_\ell . \tag{21}
$$

Again, assuming without loss of generality, that $k \in \mathbb{N}$, $\xi_k \in \{1, \dots, N\}$, using the strong convexity of $g_\xi$, the above implies for Algorithm 1 that, for all $k \geq 1$,

$$
\left\| \nabla f_{\xi_{k-1}}(x_k) - \nabla f_{\xi_{k-1}}(x_{k-1}) \right\|^2
$$
$$
= \left( g_{\xi_{k-1}}'(\langle w_{\xi_{k-1}}, x_k \rangle) - g_{\xi_{k-1}}'(\langle w_{\xi_{k-1}}, x_{k-1} \rangle) \right)^2 \left\| w_{\xi_{k-1}} \right\|^2
$$
$$
\geq \mu_{\xi_{k-1}}^2 \left( \langle w_{\xi_{k-1}}, x_k - x_{k-1} \rangle \right)^2 \left\| w_{\xi_{k-1}} \right\|^2 = \mu_{\xi_{k-1}}^2 \left( \langle w_\xi, \lambda_{k-1} \nabla f_{\xi_{k-1}}(x_{k-1}) \rangle \right)^2 \left\| w_\xi \right\|^2 .
$$

Moreover from (21), one has that

$$\left\langle w_\xi, \nabla f_{\xi_{k-1}}(x_{k-1}) \right\rangle = g'_\xi \left( w_\xi^T x \right) \| w_\xi \|^2,$$

consequently

$$\left\| \nabla f_{\xi_{k-1}}(x_k) - \nabla f_{\xi_{k-1}}(x_{k-1}) \right\|^2 \geq \mu_{\xi_{k-1}}^2 \lambda_{k-1}^2 g'_{\xi_{k-1}} \left( w_{\xi_{k-1}}^T x \right) \| w_{\xi_{k-1}} \|^6,$$

and similarly to **Case**-1, we obtain the following upper bound

$$\frac{\| x_k - x_{k-1} \|}{\left\| \nabla f_{\xi_{k-1}}(x_k) - \nabla f_{\xi_{k-1}}(x_{k-1}) \right\|} \quad \leq \quad \frac{1}{\mu_{\xi_{k-1}} \| w_{\xi_{k-1}} \|^2}.$$

From that point we repeat the same argument as in the previous section to deduce that $\sum_{k=0}^{+\infty} \lambda_k^2 < +\infty$.

# B    MISSING PROOFS AND CONVERGENCE RESULTS

We start by showing the lemmas used to control the "noisy" term in the beginning of Section 3.

## B.1    VARIANCE TRANSFER

We first recall the definition of Lipschitz continuity of the gradient.

**Definition 1.** *A function $g \colon \mathbb{R}^d \to \mathbb{R}$ is said to be locally smooth or to have locally Lipschitz continuous gradient, if $g$ is differentiable and for all $x \in \mathbb{R}^d$, there exists $L > 0$ and a neighborhood $V$ of $x$ such that, for all $x, y \in V$,*

$$\| \nabla f(x) - \nabla f(y) \| \leq L \| x - y \|.$$

*If $V = \mathbb{R}^d$, the function is simply said to be (globally) $L$-smooth or to have $L$-Lipschitz continuous gradient.*

**Lemma B.1.** *For all $x \in \mathbb{R}^d$, and random variables $\xi_1, \xi_2$ as defined in Section 1,*

$$\mathbb{E}\left[ \| \nabla f_{\xi_1}(x) - \nabla f_{\xi_2}(x) \|^2 \right]$$
$$\leq 4\mathbb{E}\left[ \| \nabla f_{\xi_1}(x) - \nabla f_{\xi_1}(x^\star) \|^2 \right] + 4\mathbb{E}\left[ \| \nabla f_{\xi_2}(x) - \nabla f_{\xi_2}(x^\star) \|^2 \right]$$
$$+ 4\mathbb{E}\left[ \| \nabla f_{\xi_1}(x^\star) \|^2 \right] + 4\mathbb{E}\left[ \| \nabla f_{\xi_2}(x^\star) \|^2 \right].$$

*Proof.* For all $x \in \mathbb{R}^d$, we have

$$\| \nabla f_{\xi_1}(x) - \nabla f_{\xi_2}(x) \|^2 \leq 2 \| \nabla f_{\xi_1}(x) \|^2 + 2 \| \nabla f_{\xi_2}(x) \|^2$$
$$= 2 \| \nabla f_{\xi_1}(x) - \nabla f_{\xi_1}(x^\star) + \nabla f_{\xi_1}(x^\star) \|^2 + 2 \| \nabla f_{\xi_2}(x) - \nabla f_{\xi_2}(x^\star) + \nabla f_{\xi_2}(x^\star) \|^2$$
$$\leq 4 \| \nabla f_{\xi_1}(x) - \nabla f_{\xi_1}(x^\star) \|^2 + 4 \| \nabla f_{\xi_1}(x^\star) \|^2 + 4 \| \nabla f_{\xi_2}(x) - \nabla f_{\xi_2}(x^\star) \|^2 + 4 \| \nabla f_{\xi_2}(x^\star) \|^2.$$

The result is then obtained by taking expectations on both sides of the above. $\square$

We can now prove Lemma 3.1.

*Proof of Lemma 3.1.* First remark that if Assumption (2-i) holds, then there is a finite number of functions $f_\xi$, according to (2). Thus, there exists a global Lipschitz constant $L = \max_{\ell \in \{1, \dots, N\}} L_\ell$ on all the $\nabla f_\xi$'s. If we rather make Assumption (2-iii), then the iterates lie in a compact subset $\mathsf{K}$ of $\mathbb{R}^d$ and each $f_\xi$ is globally[3] smooth on $\mathsf{K}$. Then by using the same argument as above, there exists a uniform Lipschitz constant $L$ on all the $\nabla f_\xi$'s. Overall, we have shown in particular that under any of the three set of assumptions, there exists $L$, such that $\forall k \geq 1$,

$$\| \nabla f_\xi(x_k) - \nabla f_\xi(x^\star) \| \leq L \| x_k - x^\star \|.$$

---

[3]This is because $\nabla f_\xi$ is locally Lipschitz continuous and $\mathsf{K}$ can be covered by a finite union of open sets.

For all $k \geq 1$, let us consider $(x_k)_{k \in \mathbb{N}}$, $(\xi_k)_{k \in \mathbb{N}}$ and $(\lambda_k)_{k \in \mathbb{N}}$ as defined in Algorithm 1. Then by direct application of Lemma B.1, we have

$$
\mathbb{E}_{k-1} \left[ \left\| \nabla f_{\xi_k}(x_k) - \nabla f_{\xi_{k-1}}(x_k) \right\|^2 \right]
$$

$$
\leq 4 \mathbb{E}_{k-1} \left[ \left\| \nabla f_{\xi_k}(x_k) - \nabla f_{\xi_k}(x^\star) \right\|^2 \right] + 4 \mathbb{E}_{k-1} \left[ \left\| \nabla f_{\xi_{k-1}}(x_k) - \nabla f_{\xi_{k-1}}(x^\star) \right\|^2 \right]
$$

$$
+ 4 \mathbb{E}_{k-1} \left[ \left\| \nabla f_{\xi_k}(x^\star) \right\|^2 \right] + 4 \mathbb{E}_{k-1} \left[ \left\| \nabla f_{\xi_{k-1}}(x^\star) \right\|^2 \right].
$$

By definition, $\sigma_k^2 = \mathbb{E}_{k-1} \left[ \left\| \nabla f_{\xi_k}(x^\star) \right\|^2 \right] + \mathbb{E}_{k-1} \left[ \left\| \nabla f_{\xi_{k-1}}(x^\star) \right\|^2 \right]$ and using the uniform Lipschitz-continuity discussed above, we obtain,

$$
\mathbb{E}_{k-1} \left[ \left\| \nabla f_{\xi_k}(x_k) - \nabla f_{\xi_{k-1}}(x_k) \right\|^2 \right] \leq 8L^2 \mathbb{E}_{k-1} \left[ \left\| x_k - x^\star \right\|^2 \right] = 8L^2 \left\| x_k - x^\star \right\|^2 + 4\sigma_k^2, \quad (22)
$$

where the last equality is obtained by noticing that $x_k$ is not random with respect to the filtration $\mathcal{F}_{k-1}$.

It only remains to prove that $\forall k \geq 1$, there exists $\sigma > 0$ such that $\sigma_k^2 \leq \sigma^2$. This is directly assumed in Assumption (2-ii). In the other-two cases, $f$ has the finite-sum structure (2). Hence, remark that in that case since $\left\| \nabla f_{\xi_k}(x^\star) \right\|^2$ does not depend on $x_k$, we can simply take the uniform bound:

$$
\sigma^2 = \max_{\ell, j \in \{1, \dots, N\}} \left\| \nabla f_\ell(x^\star) \right\|^2 + \left\| \nabla f_j(x^\star) \right\|^2.
$$

Using this in (22), we get the desired result. $\qquad \square$

## B.2 CONVERGENCE ANALYSIS

All the convergence results in Section 3 rely on the Robbins-Siegmund Theorem, a standard tool for proving convergence of random sequences. We refer the reader to (Duflo, 1997) for further discussion on this result.

**Theorem B.2** (Robbins-Siegmund). *Let $(U_k)$, $(V_k)$, $(\alpha_k)$, and $(\beta_k)$ be sequences of nonnegative and integrable random variables on some arbitrary probability space that are adapted to a filtration $\mathcal{F}_k$, and such that, almost surely,*

*(i) $\sum_{k \geq 0} \alpha_k < +\infty$ and $\sum_{k \geq 0} \beta_k < +\infty$,*

*(ii) for all $k \geq 1$,*

$$
\mathbb{E}\left[V_k | \mathcal{F}_{k-1}\right] \leq V_{k-1}(1 + \alpha_k) - U_k + \beta_k.
$$

*Then, almost surely, $V_k$ converges to a random variable $V_\infty$, and*

$$
\sum_{k \geq 0} U_k < +\infty \quad \text{almost surely.}
$$

**Proof of almost sure convergence** We will now use this theorem to prove Theorem 3.2.

*Proof of Theorem 3.2.* We use the same notations as in Section 2. According to Proposition 2.2, for Algorithm 1, for all $k \geq 2$, it holds that

$$
\mathbb{E}_{k-1}\left[T_k\right] \leq T_{k-1} + 4\lambda_k^2 \mathbb{E}_{k-1} \left[ \left\| \nabla f_{\xi_k}(x_k) - \nabla f_{\xi_{k-1}}(x_k) \right\|^2 \right],
$$

where $T_k$ is defined in (6). Then by using Assumption 2 we can apply Lemma 3.1, so there exists $L, \sigma > 0$ such that for all $k \in \mathbb{N}$,

$$
\mathbb{E}_{k-1}\left[T_k\right] \leq T_{k-1} + 32\lambda_k^2 L^2 \left\| x_k - x^\star \right\|^2 + 16\lambda_k^2 \sigma^2
$$

Notice from the definition of $T_k$ in (6), that $\left\| x_k - x^\star \right\|^2 \leq T_{k-1}$, so

$$
\mathbb{E}_{k-1}\left[T_k\right] \leq (1 + 32L^2\lambda_k^2)T_{k-1} + 16\lambda_k^2\sigma^2 \tag{23}
$$

We now apply Theorem B.2 to (23) with

$$
V_k = T_k, \ \alpha_k = 32L^2\lambda_k^2, \ \beta_k = 16\lambda_k^2\sigma^2, \text{ and } U_k = 0.
$$

Therefore, as stated in Theorem 3.2, whenever $\sum_{k \geq 0} \lambda_k^2 < +\infty$, then $\sum_{k \geq 0} \alpha_k < +\infty$ and $\sum_{k \geq 0} \beta_k < +\infty$ so we can apply Theorem B.2 to deduce that $T_k$ converges almost surely. $\qquad \square$

**Proof of convergence rates** We will now prove Theorem 3.4. We keep again the notations of Section 2 and consider Algorithm 1 with variant-III of step-size (4). This Theorem holds only for **Case**-1, which is the one where each $f_\xi$ is $\mu$-strongly convex. Importantly, this implies that $f$ is $\mu$-strongly convex as well. We begin with two technical lemmas that will be useful for proving Theorem 3.4.

**Lemma B.3.** *Let* $(\lambda_k)_{k \in \mathbb{N}}$ *denote the step-sizes of Algorithm 1 with variant-**III** of* (4). *Then under Assumption 2 and **Case**-1, for all* $k \geq 2$,

$$\frac{1}{k^{1/2+\delta}} \frac{1}{2\sqrt{2}L} \leq \lambda_k \leq \frac{1}{k^{1/2+\delta}} \frac{1}{2\sqrt{2}\mu}. \tag{24}$$

*Proof.* The upper-bound in (24) was already proved in Appendix A.1. As for the lower-bound, we proceed similarly. By applying the same arguments as in the proof of Lemma 3.1, we deduce that there exists $L > 0$ such that each $f_\xi$ has $L$-Lipschitz continuous gradient, at least on a compact set containing the iterates $(x_k)_{k \in \mathbb{N}}$. This means that

$$\left\| \nabla f_{\xi_{k-1}}(x_k) - \nabla f_{\xi_{k-1}}(x_{k-1}) \right\| \leq L \left\| x_k - x_{k-1} \right\|,$$

and thus,

$$\frac{1}{k^{1/2+\delta}} \frac{1}{2\sqrt{2}L} \leq \frac{1}{k^{1/2+\delta}} \frac{1}{2\sqrt{2}} \frac{\left\| x_k - x_{k-1} \right\|}{\left\| \nabla f_{\xi_{k-1}}(x_k) - \nabla f_{\xi_{k-1}}(x_{k-1}) \right\|}.$$

Unlike the upper-bound in (24), this is not enough to conclude. Yet, for all $k \geq 2$,

$$\lambda_k = \min \left\{ \frac{1}{k^{1/2+\delta}} \frac{\left\| x_k - x_{k-1} \right\|}{2\sqrt{2} \left\| \nabla f_{\xi_{k-1}}(x_k) - \nabla f_{\xi_{k-1}}(x_{k-1}) \right\|}, \lambda_{k-1} \sqrt{1 + \left(1 - \frac{1}{k^{1/2+\delta}}\right) \frac{\lambda_{k-1}}{\lambda_{k-2}}} \right\}$$

$$\geq \min \left\{ \frac{1}{k^{1/2+\delta}} \frac{1}{2\sqrt{2}L}, \lambda_{k-1} \sqrt{1 + \left(1 - \frac{1}{k^{1/2+\delta}}\right) \frac{\lambda_{k-1}}{\lambda_{k-2}}} \right\}$$

$$\geq \min \left\{ \frac{1}{k^{1/2+\delta}} \frac{1}{2\sqrt{2}L}, \lambda_{k-1} \right\},$$

where in the last line where used the fact that for $k \geq 2$, $\lambda_{k-1} \sqrt{1 + \left(1 - \frac{1}{k^{1/2+\delta}}\right) \frac{\lambda_{k-1}}{\lambda_{k-2}}} \geq \lambda_{k-1}$.

Therefore, by straightforward induction,

$$\lambda_k \geq \min \left\{ \frac{1}{k^{1/2+\delta}} \frac{1}{2\sqrt{2}L}, \lambda_{k-1} \right\} \geq \min \left\{ \frac{1}{k^{1/2+\delta}} \frac{1}{2\sqrt{2}L}, \frac{1}{k^{1/2+\delta}} \frac{1}{2\sqrt{2}L}, \lambda_{k-2} \right\}$$

$$\geq \ldots \geq \min \left\{ \frac{1}{k^{1/2+\delta}} \frac{1}{2\sqrt{2}L}, \lambda_1 \right\}.$$

Since $\lambda_1 = \frac{\left\| x_1 - x_0 \right\|}{2\sqrt{2} \left\| \nabla f_{\xi_0}(x_1) - \nabla f_{\xi_0}(x_0) \right\|} \geq \frac{1}{2\sqrt{2}L}$, we obtain the lower-bound in (24).

$\square$

Before presenting the next technical lemma, we recall the notation for all $k \geq 1$, $\theta_k = \lambda_k / \lambda_{k-1}$.

**Lemma B.4.** *Let* $(\lambda_k)_{k \in \mathbb{N}}$ *denotes the step-sizes of Algorithm 1 with variant-**III** of* (4). *Then under Assumption 2 and **Case**-1, for all* $k \geq 2$,

$$\frac{1}{2^\delta \sqrt{2}} \frac{\mu}{L} \leq \theta_k \tag{25}$$

$$\frac{1}{k^{1/2+\delta}} \frac{\mu}{2^\delta 4L^2} \leq \lambda_k \theta_k \leq \lambda_{k-1} (1 + \theta_{k-1}) \left(1 - \frac{1}{k^{1/2+\delta}} \frac{\mu}{\mu + 2^\delta \sqrt{2}L}\right), \tag{26}$$

*where* $\theta_k = \frac{\lambda_k}{\lambda_{k-1}}$.

*Proof.* As before we can make use of $\mu$-strong convexity and $L$-Lipschitz continuity of the gradient. In particular, using Lemma B.3, we deduce that for all $k \geq 2$,

$$\theta_k = \frac{\lambda_k}{\lambda_{k-1}} \geq \frac{(k-1)^{1/2+\delta}}{k^{1/2+\delta}} \frac{2\sqrt{2}\mu}{2\sqrt{2}L} = \left(\frac{k-1}{k}\right)^{1/2+\delta} \frac{\mu}{L}.$$

We then get (25) by noticing that for all $k \geq 2$, $\frac{k-1}{k} = 1 - \frac{1}{k} \geq \frac{1}{2}$. By using Lemma B.3 once more in (25), it holds that,

$$\lambda_k \theta_k = \frac{\lambda_k^2}{\lambda_{k-1}} \geq \frac{1}{k^{1/2+\delta}} \frac{1}{2\sqrt{2}L} \frac{1}{2^\delta\sqrt{2}} \frac{\mu}{L} = \frac{1}{k^{1/2+\delta}} \frac{\mu}{2^\delta 4L^2},$$

which is the left-hand side of (26).

Finally, for the right-hand side of (26), from (4),

$$\theta_k \lambda_k = \frac{\lambda_k^2}{\lambda_{k-1}} \leq \lambda_{k-1} \left(1 + \left(1 - \frac{1}{k^{1/2+\delta}}\right)\theta_{k-1}\right) = \lambda_{k-1}\left(1 + \theta_{k-1} - \frac{\theta_{k-1}}{k^{1/2+\delta}}\right)$$

$$= \lambda_{k-1}(1 + \theta_{k-1})\left(1 - \frac{1}{k^{1/2+\delta}} \frac{\theta_{k-1}}{1+\theta_{k-1}}\right).$$

One can easily show that the function $\theta \in \mathbb{R} \mapsto \frac{\theta}{1+\theta}$ is increasing on $\mathbb{R}$. So combining this with (25) again,

$$\frac{\theta_{k-1}}{1+\theta_{k-1}} \geq \frac{\frac{1}{2^\delta\sqrt{2}}\frac{\mu}{L}}{1 + \frac{1}{2^\delta\sqrt{2}}\frac{\mu}{L}} = \frac{\mu}{2^\delta\sqrt{2}L + \mu}.$$

So

$$1 - \frac{1}{k^{1/2+\delta}} \frac{\theta_{k-1}}{1+\theta_{k-1}} \leq 1 - \frac{1}{k^{1/2+\delta}} \frac{\mu}{2^\delta\sqrt{2}L + \mu},$$

which proves the last bound. $\qquad\square$

We can now prove Theorem 3.4.

*Proof of Theorem 3.4.* As before, let us consider $(x_k)_{k\in\mathbb{N}}$ the iterates of Algorithm 1 with step-sizes $(\lambda_k)_{k\in\mathbb{N}}$ obtained with variant-**III**. With the same arguments as above, upon restriction to a compact set, all $\nabla f_\xi$ and $\nabla f$ are $L$-Lipschitz continuous due to Assumption 2. In **Case**-1, $f$ and the $f_\xi$'s are furthermore $\mu$-strongly convex.

Under these stronger assumptions we can actually reproduce the steps of the analysis from Section 2 but with sharper bounds. Indeed, we can go back to the convexity inequality (8) and replace it with

$$-2\langle \mathbb{E}_{k-1}[\lambda_k \nabla f_{\xi_k}(x_k)], x_k - x^\star\rangle = -2\lambda_k \langle \nabla f(x_k), x_k - x^\star\rangle$$
$$\leq -2\lambda_k(f(x_k) - f^\star) - \lambda_k\mu\|x_k - x^\star\|,$$

where we also dropped the expectations on $\lambda_k$ as our choice is not random conditionally on $\mathcal{F}_{k-1}$. We apply the same reasoning to (10): strong convexity implies that

$$f(x_{k-1}) \geq f(x_k) - \langle x_k - x_{k-1}, \nabla f(x_k)\rangle + \frac{\mu}{2}\|x_k - x_{k-1}\|^2$$

and hence going back to (10),

$$\mathbb{E}_{k-1}[B_k] \leq 2\frac{\lambda_k^2}{\lambda_{k-1}}(f(x_{k-1}) - f(x_k)) - 2\frac{\lambda_k^2}{\lambda_{k-1}}\mu\|x_k - x_{k-1}\|^2$$

$$= 2\frac{\lambda_k^2}{\lambda_{k-1}}(f(x_{k-1}) - f^\star) - 2\frac{\lambda_k^2}{\lambda_{k-1}}(f(x_k) - f^\star) - \frac{\lambda_k^2}{\lambda_{k-1}}\mu\|x_k - x_{k-1}\|^2.$$

We then proceed exactly as in Section 2, and eventually obtain a sharper bound on $\mathbb{E}_{k-1}[T_k]$:

$$\mathbb{E}_{k-1}[T_k] \leq (1 - \lambda_k\mu)\|x_k - x^\star\|^2 + 2\lambda_k\theta_k(f(x_{k-1}) - f^\star) + (1 - 2\lambda_k\theta_k\mu)\frac{\|\Delta_{k-1}\|^2}{2}$$

$$+ 4\lambda_k^2 \mathbb{E}_{k-1}\left[\|\nabla f_{\xi_k}(x_k) - \nabla f_{\xi_{k-1}}(x_k)\|^2\right], \tag{27}$$

where we recall that $\Delta_{k-1} = \|x_k - x_{k-1}\|$.

We now use Lemma B.3 and B.4, more specifically, we use the left-hand side of (24), the right-hand side of (26) and the left-hand side of (26) on the first, second and third terms of (27), respectively. This yields for all $k \geq 2$,

$$
\mathbb{E}_{k-1}[T_k] \leq \left(1 - \frac{1}{k^{1/2+\delta}} \frac{\mu}{2\sqrt{2}L}\right) \|x_k - x^\star\|^2
$$
$$
+ 2(1 + \theta_{k-1}) \left(1 - \frac{1}{k^{1/2+\delta}} \frac{\mu}{\mu + 2^\delta \sqrt{2}L}\right) (f(x_{k-1}) - f^\star)
$$
$$
+ \left(1 - \frac{1}{k^{1/2+\delta}} \frac{\mu^2}{2^\delta 2L^2}\right) \frac{\|\Delta_{k-1}\|^2}{2}
$$
$$
+ 4\lambda_k^2 \mathbb{E}_{k-1}\left[\left\|\nabla f_{\xi_k}(x_k) - \nabla f_{\xi_{k-1}}(x_k)\right\|^2\right],
$$

Let $\tau = \min\left\{\frac{\mu}{2\sqrt{2}L}, \frac{\mu}{\mu+2^\delta\sqrt{2}L}, \frac{\mu^2}{2^\delta 2L^2}\right\} < 1$, because $\mu/L \leq 1$. and so we obtain an improved version of Proposition 2.2: for all $k \geq 2$,

$$
\mathbb{E}_{k-1}[T_k] \leq \left(1 - \frac{\tau}{k^{1/2+\delta}}\right) + 4\lambda_k^2 \mathbb{E}_{k-1}\left[\left\|\nabla f_{\xi_k}(x_k) - \nabla f_{\xi_{k-1}}(x_k)\right\|^2\right],
$$

It now remains to control the last term above, in the exact same way the proof of Theorem 3.2. We apply Lemma 3.1 to get

$$
\mathbb{E}_{k-1}[T_k] \leq \left(1 - \frac{\tau}{k^{1/2+\delta}}\right) T_{k-1} + 32L\lambda_k^2 \|x_k - x^\star\| + 16\lambda_k^2 \sigma^2,
$$
$$
\leq \left(1 - \frac{\tau}{k^{1/2+\delta}} + 32L\lambda_k^2\right) T_{k-1} + 16\lambda_k^2 \sigma^2.
$$
$$
\leq \left(1 - \frac{\tau}{k^{1/2+\delta}} + \frac{1}{k^{1+2\delta}} \frac{32L}{8\mu^2}\right) T_{k-1} + 16\lambda_k^2 \sigma^2.
$$

There exists $k_0$ such that the factor of $T_{k-1}$ is strictly smaller than 1. For example, it is sufficient to take $k_0 = \frac{16L^2}{\tau^2 \mu^4}$. Using the fact that $\lambda_k^2$ is summable, we can use (Bercu & Bigot, 2021, Lemma A.3) to deduce the result. $\qquad\square$

## C  THE CONVEX CASE

In this section, we consider a relaxation of the smooth and strongly convex case that we considered previously. Let us recall that, from (15) in Proposition 2.2, we have:

$$
\mathbb{E}_{k-1}\left[\|x_{k+1} - x^\star\|^2 + \frac{\|x_{k+1} - x_k\|^2}{2}\right]
$$

$$
\leq \underbrace{\|x_k - x^\star\|^2 + \frac{\|x_k - x_{k-1}\|^2}{2}}_{:=V_k} + \underbrace{4\mathbb{E}_{k-1}\left[\lambda_k^2\right] \mathbb{E}_k\left[\left\|\nabla f_{\xi_k}(x_k) - \nabla f_{\xi_{k-1}}(x_k)\right\|^2\right]}_{:=\beta_k}
$$
$$
\underbrace{+ 2\frac{\mathbb{E}_{k-1}\left[\lambda_k^2\right]}{\lambda_{k-1}} (f(x_{k-1}) - f^\star) - 2\left(\mathbb{E}_{k-1}[\lambda_k] + \frac{\mathbb{E}_{k-1}\left[\lambda_k^2\right]}{\lambda_{k-1}}\right) (f(x_k) - f^\star)}_{:=-U_k} \qquad (28)
$$

From now on, we assume that the step-size $\lambda_k$ satisfies the following update rule:

$$
\lambda_k = \min\left\{\frac{\|x_k-x_{k-1}\|}{2\sqrt{2}\|\nabla f_{\xi_{k-1}}(x_k)-\nabla f_{\xi_{k-1}}(x_{k-1})\|}, \lambda_{k-1}\sqrt{1+\theta_{k-1}}, \right.
$$
$$
\left. \frac{f(x_k)-f^\star}{f(x_{k-1})-f(x_k)}\lambda_{k-1}\mathbf{1}_{\{f(x_{k-1})-f(x_k)>0\}}\right\}. \qquad (29)
$$

Remark that with choice (4), $\lambda_k$ is not a random variable conditionally on $\mathcal{F}_{k-1}$, so we simply have $\mathbb{E}_{k-1}[\lambda_k] = \lambda_k$ and $\mathbb{E}_{k-1}\left[\lambda_k^2\right] = \lambda_k^2$ in (28).

**Remark:** One should notice that the step-size rule is different in this last part of the paper, with an additional third condition to ensure the non-negativeness of $U_k$. As for the second condition, we chose the simplest one, coming from Malitsky & Mishchenko (2020), but we could of course use stronger conditions such as in previous subsections.

Since we do no longer make a strong convexity assumption, we will need the next one instead.

**Assumption 3** (Square Summability of the step-sizes). *The sequence $(\lambda_k)_{k \in \mathbb{N}}$ of step-sizes satisfies*

$$\sum_{k \geq 0} \lambda_k^2 < +\infty.$$

**Lemma C.1.** *Suppose that Assumption 2 holds, and that Assumption 3 is satisfied for the choice of step-size* (29). *Then, using the notations of Equation (28) we have that*

$$\mathbb{E}_{k-1}[V_k] \leq V_{k-1} - U_k + \beta_k$$

*with $\sum_{k \geq 0} \beta_k < +\infty$, and $U_k \geq 0$.*

*Proof.* Thanks to Assumption 2 and the hypothesis that $\lambda_k$ satisfies Assumption 3, we have that $\sum_{k \geq 0} \beta_k < +\infty$. Now let us consider $U_k$ to check when it is non-negative:

$$U_k = -2 \frac{\lambda_k^2}{\lambda_{k-1}} \left( f(x_{k-1}) - f^\star \right) + 2 \left( \lambda_k + \frac{\lambda_k^2}{\lambda_{k-1}} \right) \left( f(x_k) - f^\star \right),$$

that is

$$U_k = 2 \frac{\lambda_k^2}{\lambda_{k-1}} \left( f(x_k) - f(x_{k-1}) \right) + 2\lambda_k \left( f(x_k) - f^\star \right)$$

Notice that we always have $f(x_k) - f^\star \geq 0$. Two cases may occur:

  (a) If $f(x_k) \geq f(x_{k-1})$, then we always have $U_k \geq 0$.

  (b) If $f(x_{k-1}) > f(x_k)$, then we see that $U_k \geq 0$ if and only if

$$\lambda_k \leq \frac{f(x_k) - f^\star}{f(x_{k-1}) - f(x_k)} \lambda_{k-1},$$

which concludes the proof thanks to the choice (29) for $\lambda_k$.

$\square$

**Proposition C.2.** *Suppose that Assumptions 2 holds, and that Assumption 3 is satisfied for the choice of step-size* (29). *Then, we have that*

$$\sup_{k \geq 0} \mathbb{E}\left[ \|x_k - x^*\| \right] < +\infty$$

*Proof.* Thanks to Lemma C.1, we can apply the Robbins-Siegmund theorem (Theorem B.2) with $V_k = \|x_k - x^\star\|^2 + \frac{\|x_k - x_{k-1}\|^2}{2}$ which concludes the proof. $\square$

We can now give convergence results.

**Proposition C.3.** *Let us define*

$$\hat{x}_K = \frac{\left( \lambda_K + \frac{\lambda_K^2}{\lambda_{K-1}} \right) x_K + \sum_{k=1}^{K-1} w_k x_k}{S_K}$$

*with*

$$w_k = \lambda_k + \frac{\lambda_k^2}{\lambda_{k-1}} - \frac{\lambda_{k+1}^2}{\lambda_k}$$

*and*

$$S_K = \lambda_K + \frac{\lambda_K^2}{\lambda_{K-1}} + \sum_{k=1}^{K-1} w_k = \frac{\lambda_1^2}{\lambda_0} + \sum_{k=1}^{K} \lambda_k$$

*Suppose that Assumption 2 holds, and that Assumption 3 is satisfied for the choice of step-size* (29). *Then, there exists $C > 0$ such that we have*

$$f(\hat{x}_K) - f^* \leq \frac{C}{S_K} \text{ a.s.}$$

*Proof.* We have:

$$\sum_{k=1}^{K} U_k = 2 \left( \lambda_K + \frac{\lambda_K^2}{\lambda_{K-1}} \right) (f(x_K) - f^\star) - 2 \frac{\mathbb{E}_0 \left[ \lambda_1^2 \right]}{\lambda_0} (f(x_0) - f^\star)$$

$$+ 2 \sum_{k=1}^{K-1} \left( \underbrace{\left( \lambda_k + \frac{\lambda_k^2}{\lambda_{k-1}} - \frac{\lambda_{k+1}^2}{\lambda_k} \right)}_{= w_k} (f(x_k) - f^\star) \right)$$

Thanks to Lemma C.1, we can apply Robbins-Siegmund theorem (Theorem B.2) to deduce that $\sum_{k \geq 0} U_k < +\infty$ a.s. Then, notice that from the update rule (29) of $\lambda_k$, we have that $w_k \geq 0$. We thus deduce that there exists a constant $C > 0$ such that:

$$\left( \lambda_K + \frac{\lambda_K^2}{\lambda_{K-1}} \right) (f(x_K) - f^\star) + \sum_{k=1}^{K-1} w_k (f(x_k) - f^\star) \leq C \text{ a.s.}$$

Since $f$ is convex, we now want to use Jensen inequality, that states that if $\sum_{k=1}^{K} \rho_k = 1$:

$$f \left( \sum_{k=1}^{K} \rho_k x_k \right) \leq \sum_{k=1}^{K} \rho_k f(x_k)$$

So that we get

$$f(\hat{x}_K) - f^* \leq \frac{C}{S_K} \text{ a.s.} \tag{30}$$

which concludes the proof. $\qquad\square$

We finally add a last assumption.

**Assumption 4** (Divergence of the steps)**.** *The random sequence $(\lambda_k)_{k \in \mathbb{N}}$ of step-sizes satisfies*

$$\sum_{k \geq 0} \lambda_k = +\infty$$

Notice that Assumption 4 is satisfied in the smooth and strongly convex case. Notice also that this assumption would be directly satisfied if we only had the two first conditions in the update rule of $\lambda_k$.

**Corollary C.4.** *Suppose that Assumptions 2 holds, and that Assumption 3 and 4 are satisfied for the choice of step-size* (29). *Then, we have that*

$$f(\hat{x}_K) - f^* \to 0 \text{ a.s.}$$

*Proof.* Notice that

$$S_K = \frac{\lambda_1^2}{\lambda_0} + \sum_{k=1}^{K} \lambda_k \geq \sum_{k=1}^{K} \lambda_k$$

Hence, by using Assumption 4 in (30), we get the result of the corollary.

$\qquad\square$

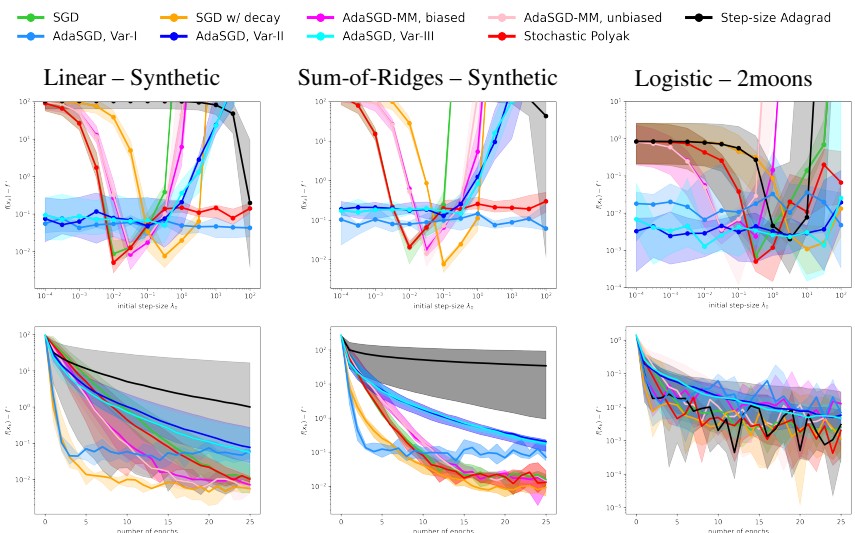

Figure 4: Sensitivity to $\lambda_0$ and minimization performance after tuning for **smaller** mini-batches (of size 8). Same experimental setting as in Figures 1, 2 and 3.

## D DETAILS ON THE EXPERIMENTS

**Objective functions.** We performed numerical experiments on problems of the form (2) *i.e.,* where $f = \frac{1}{N} \sum_{\ell=1}^{N} f_\ell$. We considered four types of loss functions defined respectively for all $x \in \mathbb{R}^d$ by:

- Linear regression: $f_\ell(x) = \frac{1}{2}(y_\ell - w_\ell^T x)^2$

- Ridge regression: $f_\ell(x) = g_\ell\left(y_\ell - w_\ell^T x\right)$ where $g_\ell(t) = \frac{t^4}{1+t^2} + 10^{-2}t^2$ is $10^{-2}$-strongly convex.

- Logistic regression: $f_\ell(x) = \log\left(1 + \exp(-y_\ell w_\ell^T x)\right)$

- Poisson regression: $f_\ell(x) = \log\left(-y_\ell w_\ell^T x + 1 + \exp(w_\ell^T x)\right)$

In the above, the dataset is made of couples $(w_\ell, y_\ell) \in \mathbb{R}^d \times \mathbb{R}$ for $\ell \in \{1, \ldots, N\}$. Below we describe the data as a matrix $W \in \mathbb{R}^{N \times d}$ and a vector $y \in \mathbb{R}^N$.

**Datasets** Based on the above, we used synthetic and real-world datasets of the form $(W, y)$ with $W \in \mathbb{R}^{N \times d}$ and $y \in \mathbb{R}^N$, where $N$ is the number of data samples.

- Synthetic datasets were generated by drawing data at random. Each element of $W$ is i.i.d. $\mathcal{N}(0, 1)$. We do the same for the vector $y$. We used $N = 200$ and $d = 20$. We used batch-size of 32 (*i.e.,* $\text{Card}(\xi_k) = 32$).

- The *2moons* dataset is available via the scikit learn library (Pedregosa et al., 2011). It is made of $N = 200$ samples in dimension $d = 3$. We used a batch-size of 32.

- The *w8a* dataset[4] is a publicly available dataset classically used for logistic regression. It has a significantly larger number $N = 49749$ of samples, and is also larger-dimensional as $d = 300$. We used a batchsize of 309.

All the algorithms described below are ran for the same number of epochs, where one epoch consists in using $N$ data samples. That is, one epoch corresponds to $\frac{N}{\text{batch}-\text{size}}$ iterations. At each iteration, we sample the mini-batches of data $(\xi_k)_{k \in \mathbb{N}}$ uniformly at random with replacement.

---

[4]https://www.csie.ntu.edu.tw/~cjlin/libsvmtools/datasets/binary.html

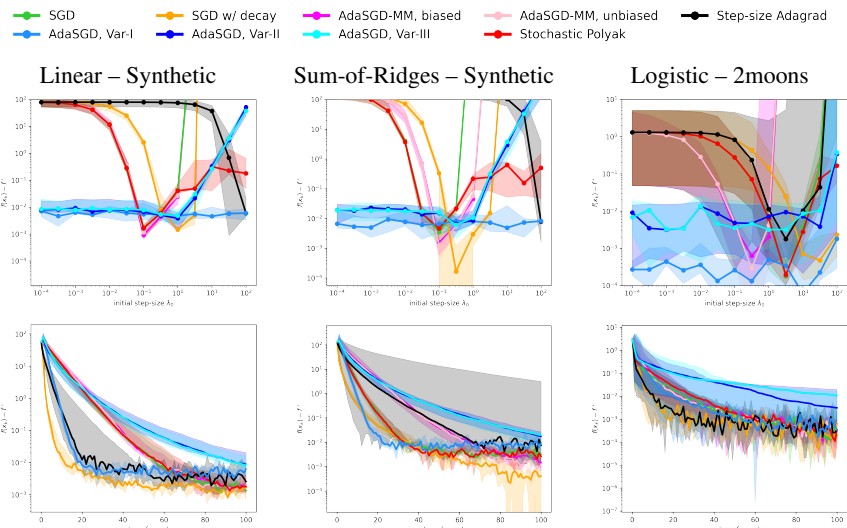

Figure 5: Sensitivity to $\lambda_0$ and minimization performance after tuning for **larger** mini-batches (of size 128). Same experimental setting as in Figures 1, 2 and 3.

**Algorithms**

- SGD is the algorithm described in (3). It is either used with no decay ($\lambda_k = \lambda_0$) or decay $\lambda_k = \frac{\lambda_0}{k^{1/2+\delta}}$.

- We implemented the three variants of AdaSGD exactly as described in Algorithm 1 and (4).

- AdaSGD-MM is implemented as proposed by Malitsky & Mishchenko (2020). As previously discussed, it is SGD with a step-size that is computed for $k \geq 2$ as $\lambda_k = \min \left\{ \alpha \Lambda_k, \lambda_{k-1}\sqrt{1+\theta_{k-1}} \right\}$ with either $\Lambda_k = \frac{\|x_k - x_{k-1}\|}{\|\nabla f_{\xi_k}(x_k) - \nabla f_{\xi_k}(x_{k-1})\|}$ (referred to as biased) or $\Lambda_k = \frac{\|x_k - x_{k-1}\|}{\|\nabla f_{\zeta_k}(x_k) - \nabla f_{\zeta_k}(x_{k-1})\|}$ (unbiased), where $\zeta_k$ is an independent copy of $\xi_k$, and where $\alpha > 0$. To ease the discussion and comparison, in Section 4, we call $\lambda_0$ this parameter $\alpha$ as it is the tunable parameter of the method.

- Step-size Adagrad uses the step-size $\lambda_k = \frac{\lambda_0}{\sqrt{v_k}}$ where $v_k$ is updated iteratively: $v_k = v_{k-1} + \|\nabla f_{\xi_k}(x_k)\|^2$, starting with $v_{-1} = 0$.

- The stochastic Polyak-stepsize is defined by $\lambda_k = \min \left( \frac{f_{\xi_k}(x_k) - f_{\xi_k}^\star}{\|\nabla f_{\xi_k}(x_k)\|^2}, \lambda_0 \right)$. It is important to note that this step-size rule additionally requires evaluating $f_{\xi_k}(x_k)$ at each iteration and is far from being tuning free. Indeed, it features a step-size $\lambda_0$ that needs to be tuned but also assumes knowledge of all the optimal values $f_\ell, n \in \{1, \ldots, N\}$. For fair comparisons with other methods, here we replace the optimal value $f_\ell^\star$ by 0 for all $n$, since one cannot assume to know these values *a priori*.

**Parameter selection.**

- All the methods feature a single tunable parameter $\lambda_0$. To obtain Figures 1 and 2, we ran the algorithms described above for all choices $\lambda_0$ on a dense grid of values. More specifically, we tried every $\lambda_0$ of the form $10^i$, for $i \in \{-4, -3.5, \ldots, 1.5, 2\}$. We did 100 epochs for each problem, except for the *w8a* dataset for which we did the grid-search on 10 epochs due to a significantly higher computational cost.

- For Figure 3, and bottom-rows of Figures 4 and 5, we selected the parameter $\lambda_0$ that achieved the lowest full-batch value $f(x_k)$ at the last epoch of the grid-search. We then

ran again the algorithm with the best choice of $\lambda_0$ observed in the above experiments. We proceeded this way for all the baselines (SGD and AdaSGD-MM), however, since we argue that AdaSGD requires no tuning, we did not select $\lambda_0$ for our methods (AdaSGD), and rather always used a step-size $\lambda_0 = 10^{-3}$.

- For SGD with decay, as well as Variants-**II** and **III** of AdaSGD, we use $\delta = 10^{-2}$. This value was *not tuned* and simply taken small as any small $\delta > 0$ works from a theoretical point of view.

