# OpenReview forum: "Stochastic Adaptive Gradient Descent Without Descent"
_ICLR.cc/2026/Conference — Submitted to ICLR 2026_

### Official Review · Reviewer_8tA4 · 2025-10-25

**Soundness:** 3
**Presentation:** 2
**Contribution:** 1
**Rating:** 4
**Confidence:** 3

**Summary:**

This paper investigate Adaptive Gradient Descent without Descent method to the stochastic convex optimization setting. It proposes multiple adaptive step-size strategies that preserve the desirable properties of adaptive methods in the deterministic case. The authors also conduct experiments to validate the effectiveness of their proposed approach.

**Strengths:**

- This paper is well written and easy to follow. The motivation is clearly presented and well explained.
- The authors provide experimental results to support their claims.

**Weaknesses:**

- My primary concern lies in the contribution and the novelty of this paper. The core idea of this paper is the decomposition (11).  Could the authors summarize the key technical challenges in their proof?
- In Assumption (2-iii), the authors assume $x_k$ is bounded. Why not simply introduce a projection operator to ensure boundedness instead?
- In Appendix C, it is unclear what the order of $S_K$. The authors should explicitly state or derive convergence rate respect to $k$.
- The authors should also compare different batch sizes in their experiments to better illustrate the scalability and robustness of their method.

**Questions:**

Please refer to __Weakness__.

**Details Of Ethics Concerns:**

None.

---

> ### Author Response · Authors · 2025-11-20
>
> ## On the contribution and Novelty of the paper
> First, we believe that successfully adapting the method of [Malitsky and Mishchenko, 2019] to the stochastic setting is an important contribution on its own. The method has indeed received a lot of attention in the deterministic setting, but had not yet been as successfully adapted to the stochastic case (please see also our answer to Reviewer CXgc on this point).
>
> On the technical side, while Equation~11 is an important piece of our derivations, there are several technical challenges to overcome before getting there. Here are some examples:
> 1. One needs to ensure $\lambda_k$ remains conditionnally independant of $\xi_k$, while maintaining a reasonnable computational cost. Already one of the two stochastic adaptations of [Malitsky and Mishchenko, 2019] fails to provide this.
> 2. In Equation~9 we add and substract the term $\nabla f_{\xi_{k-1}}(x_{k-1})$. There are many possible choices at this stage (using $\xi_k$, $\xi_{k-1}$, $x_k$, $x_{k-1}$, etc.). Only our choice allows getting the two terms $A_k$ and $B_k$ that later allows us to obtain a method that is the right adaptation to the stochastic setting (as it boils down to the deterministic one when taking exact gradients, see the last paragraph of Section 2).
>
> These is exactly because we found the correct arguments and put them in the correct order that the proof may look "simple" at first sight.
>
> ## On the experiments
> We added additional experiments to the new version of the paper with small and large batch-sizes as advised by the reviewer. They show that our adaptive step-size methods remain stable to the choice of $\lambda_0$. We also improved significantly the experimental section (please see the paper and our answers to other reviewers).
>
> ## Other remarks
> * **About $S_K$**  The reviewer is correct about $S_K$ in Appendix C. This result is merely an extension that we added to give some perspectives for future work on the general convex case, but is not the main focus of the current work.
>
> * **On Assumption (2-iii)**
> The boundedness in Assumption (2-iii) is not restrictive in the sense that it needs not be true.
> We stress that Assumptions (i), (ii), (iii), are *stand-alone*, only one needs to be true, therefore (iii) is not necessary for the results to hold true. Adding a projection is not possible as would mean knowing a set that contains $x^\star$.

---

### Official Review · Reviewer_mtrd · 2025-10-29

**Soundness:** 3
**Presentation:** 3
**Contribution:** 2
**Rating:** 6
**Confidence:** 3

**Summary:**

The paper proposes a new adaptive step-size strategy for stochastic gradient methods for strongly convex optimization problems. The approach adapts the step size based solely on first-order information, without requiring any problem-specific parameters about the level of convexity and smoothness, that are typically not known, or manual tuning. It is derived from a theoretical extension of the Adaptive Gradient Descent Without Descent method to the stochastic setting. The paper provide convergence guarantees under several assumptions and demonstrate through experiments that the method performs well, especially, that it is less sensitive to initial step-size selection than some other step-size strategies for SGD.

**Strengths:**

The paper studies on adaptive step-size selection, which addresses a critical aspects of stochastic optimization. Step-size choice strongly influences both convergence speed and stability, and many existing work that establish convergence do so by tuning the step-size to problem parameters such as the level of convexity or smoothness, which are rarely known in advance. The proposed method offers a valuable contribution by automatically adapting the step size to the local geometry of the objective function in an online manner, without requiring prior information or manual tuning. This data-driven adaptivity enhances the method’s practicality and robustness, making it potentially useful for a wide range of real-world optimization problems.

The experiment results make a good case for how the adaptive approach is less sensitive to hyperparameters. The baseline algorithms seem reasonably well justified, though, I think for strongly convex problems O(1/t) step-size should be better than O(1/sqrt(t)) stepsize.

The theoretical development is clear and rigorous. The authors provide a well-motivated adaptive rule and prove convergence under standard assumptions, which strengthens the overall credibility of the approach.

The paper makes a useful contribution by extending adaptive step-size ideas, previously explored in deterministic settings, to the stochastic regime. This connection to existing theory is well-articulated and strengthens the paper’s motivation.

**Weaknesses:**

From a theory standpoint, I do not see a clear advantage of the proposed adaptive step-size scheme. While I acknowledge that adaptivity can be beneficial in practice by removing the need to know parameters such as the Lipschitz constant L or the strong convexity parameter \mu, this does not appear to yield any improvement in theoretical convergence guarantees. In particular, you don't improve theoretical performance for, e.g., O(1/t) step-size that does not have knowledge of L or \mu.

In general, knowledge of the Lipschitz constant is not required to ensure convergence or even a convergence rate in either stochastic or deterministic optimization; both can achieve convergence with appropriately diminishing step sizes. In deterministic optimization, however, diminishing step sizes are typically suboptimal, as they result in slower (sublinear) convergence. In such cases, using a constant step size proportional to 1/L provides improved linear convergence, which motivates adaptive schemes such as Malitsky & Mishchenko (2020) that effectively approximate 1/L without requiring prior knowledge of L.

In contrast, in stochastic optimization, standard diminishing step sizes such as α_t = O(1/t) already achieve O(1/t) convergence rate in the strongly convex and L-smooth setting, without requiring knowledge of L or \mu. Therefore, from a theoretical standpoint, adapting the step size to local curvature does not improve the convergence rate. The main benefit of such adaptive step-size rules would thus be in practice, potentially improving empirical stability or constant factors, rather than providing stronger theoretical guarantees.

Another limitation of the experimental results is the lack of statistical analysis or evidence of robustness across multiple runs. The reported results appear to be based on a single execution, which makes it difficult to assess the variability and reliability of the observed performance. Including averages and standard deviations over several independent runs would strengthen the empirical evaluation and support the claimed advantages of the proposed method.

**Questions:**

I feel Figure 1 is not so convincing because there is much information lacking. For example, how long do you run the algorithms? Is it always the same number of steps? How are the step-sizes selected for other algorithms? Is this from a single run? How do you know you were not just lucky?

Why do you only consider strongly convexed functions? Is the idea limited to convexity? Or is it easy to extend to non-convex problems?

---

> ### Author Response · Authors · 2025-11-20
>
> ## On the main concern that rates of convergence are not improved
> Our adaptive step-size does not improve upon the rate of vanilla SGD. In fact, this was also true for the method by [Malitsky and Mishchenko, 2019] in the deterministic setting. We would like to clarify that our main focus is removing the need of tuning the step-size which is a crucial difficulty for stochastic algorithms (please see also our answer to Reviewer CXgc).
>
> ## On asymptotic convergence for Vanilla-SGD without knowledge of $L$
> The reviewer is correct that **in theory**, for SGD with deminishing step-sizes, one can get convergences without knowning the smoothness constant $L$.
> Informally, this is because if $\lambda_k = \frac{\lambda_0}{k}$, then there exists $k_0$ such that $\frac{\lambda_0}{k} \leq \frac{1}{L}$, that is $k_0 = \lambda_0 L$.
> Yet, we would like to point out a crucial difference in our results, that is that the iteration after which our rate holds true **is independant of $\lambda_0$**. This was mentionned after Theorem 3.4, but we now added an extended remark (Remark 3.5) to clarify this key difference. So the improvement is not only empirical.
>
> Furthermore, to get the optimal rate for SGD in strongly convex optimization, [Bach and Moulines, 2011](https://papers.nips.cc/paper_files/paper/2011/hash/40008b9a5380fcacce3976bf7c08af5b-Abstract.html) showed that the knowledge of the strong-convexity parameter $\mu$ is crucial (theoretically and empirically) to adjust the step-size for vanilla SGD (see Section 3.1 ``Behavior for $\alpha=1$'' in the aforementioned paper). These considerations are also asymptotic, but in practice the choice of $\lambda_0$ has a significant impact on the performance of vanilla SGD as shown in Section 6 of [Asi and Duchi, 2018](https://arxiv.org/pdf/1810.05633).
>
> To summarize finding the right initial step-size is crucial for vanilla SGD, in particular for non-asymptotic performance. We believe that this makes the need for non-asymptotic adaptive strategies (such as our algorithm) particularly relevant.
>
> ## On strong-convexity
> We would like to clarify that we do not only consider strongly-convex functions. We also consider cases (Case-2 and Case-3 in the paper), where functions admit only one directionally strongly convex, but **might have degenerate Hessian in all other directions**, which is far from strong convexity.
>
> ## On Figure 1
> We now ran each experiments five times and reported median, worst and best runs (please see the new version, as well as our answer to Reviewer CXgc). The new experiments show that the stability the tuning-free nature of our method is consistent across runs of the same experiments (and across problems).

---

### Official Review · Reviewer_CXgc · 2025-10-31

**Soundness:** 3
**Presentation:** 3
**Contribution:** 2
**Rating:** 4
**Confidence:** 3

**Summary:**

The paper proposes AdaSGD, a parameter-free adaptive step-size strategy for stochastic convex optimization that extends “Adaptive Gradient Descent Without Descent” (AdaGD) (a previous paper) to the stochastic setting. The key idea is to estimate a local curvature proxy using the previous mini-batch, yielding three practical variants of an adaptive rule (Eq. (4)) that do not require tuning any problem-dependent scale (apart from a theory-only $\epsilon$-like exponent $\delta$). The method uses a Lyapunov construction adapted from the deterministic analysis and proves (i) almost-sure convergence under standard assumptions via Robbins–Siegmund and (ii) a rate $\mathbb{E}\|x_{k+1}-x^*\|^2\leq C/k^{0.5+\delta}$ for strongly convex objectives (Variant III). Empirically, on linear, ridge, logistic, and Poisson regression, AdaSGD is markedly robust to the initial step size and performs comparably to tuned baselines, while requiring two gradient evaluations per iteration.

**Strengths:**

The paper offers a clear and principled stochastic adaptation of the deterministic AdaGD scheme by computing the step size from the previous mini-batch, yielding three concrete variants in Eq. (4) and a concise implementation in Algorithm 1. The analysis is careful: a Lyapunov function (Eq. (6)) is constructed to obtain a one-step descent-in-expectation inequality, leading to almost-sure convergence under Assumption 2 when the steps are square-summable (Theorem 3.2), and to an explicit rate $\mathbb{E}\|x_{k+1}-x^*\|^2\leq C/k^{0.5+\delta}$ in the strongly convex case for Variant III (Theorem 3.4).

The experimental section (Sec. 4) is focused and reproducible: objectives (linear, ridge, logistic, Poisson) are written explicitly, datasets are described with sizes and batch settings and the evaluation protocol uses epochs and mini-batch sampling with replacement. Figures 1 and 2 convincingly show AdaSGD attains competitive sub-optimality across problems.

The paper is generally well-written and organized; the step-size rule is easy to implement, and the connection to the deterministic method is explained clearly in Sec. 2.

To sum up:
1. The paper has solid mathematical analysis with clear assumptions, statements and proofs.
2. The writing is generally clear with nice flow.
3. The work includes experimental work that compares the proposed method with other works.

**Weaknesses:**

The empirical scope is narrow and lacks statistical rigor: results are reported on a small suite of convex problems (with one larger dataset, w8a), but there are no error bars, multiple seeds, or variance measures in Figs. 1-3, making it hard to assess significance. Furthermore, baseline coverage is limited: while SGD (with and without decay) and AdaSGD-MM are included, widely used parameter-free/adaptive competitors (such as Adagrad/Adam/Polyak Stepsizes) are not evaluated in Sec. 4, so the practical advantage over common alternatives remains unclear.

Furthermore, on the practicality side, the per-iteration cost is higher due to the second gradient evaluation, although the paper argues this is offset by avoiding grid search; there is no wall-clock comparison to prove that claim.

However, my main concern with this work is the significance/originality. As the paper clearly mentions, this work extends the deterministic results of a previous paper to the stochastic setting. I appreciate the techniques used in order to achieve that, but it is not clear to me what is the *significance* of this extension.

**Questions:**

See weaknesses.

To sum up:
1. Could you add empirical comparisons with a few parameter-free/adaptive baselines mentioned in the related work on the same convex tasks?
2. Can you include wall-clock/time-to-target comparisons that account for the second gradient evaluation per iteration and contrast them with typical hyperparameter grid searches for SGD?
3. Can you explain why the extension to the stochastic regime is important? Is it a theoretical benefit? Is it a practical benefit?

---

> ### Author Response · Authors · 2025-11-20
>
> ## On error bars
> We have now rerun all experiments five times, and updated the plots to show the median, worst, and best run for each algorithm.
> Except for a few outliers, the conclusions remain the same as in the initial version of our paper: AdaSGD is far more stable to the initial $\lambda_0$ than other method. We thank the reviewer for suggesting these additional runs that  strenghten our empirical evaluations (see Figures 1, 2 and 3).
>
> ## On additional adaptive baselines
> While our primary focus was to showcase the improvment compared to the original stochastic version of AdaGD, we agree that it is interesting to compare to additional baseline.
> To this aim we now uploaded all figures to add  comparison with two other adptive step-sizes: [Stochastic Polyak Step-size](https://proceedings.mlr.press/v130/loizou21a/loizou21a.pdf) (SPS) and step-size Adagrad (s-Adagrad).
> These two methods actually feature "step-size-like" hyper-parameters that we tuned. The conclusion is that the tuning of this parameter is important for both new baselines, much more than for our AdaSGD algorithm (see Figures 1 and 2).
> ## On significance of our work
>
> Stochastic optimization is of uttermost importance, especially nowadays, for training machine learning models on very large datasets where full-batch evaluations are unpractical.
> Managing to adapt a deterministic method is not a little improvement as the stochastic setting anihilates most adaptive methods available in the deterministic setting (line-search, doubling trick, etc., see Section 1.3) which makes the tuning of step-sizes significantly more involved. To this day this remains a hardship in machine learning with numerous papers on the topic.
>
> The work [Malitsky and Mishchenko, 2019] represents a powerful alternative to standard approaches in the deterministic setting but was not "tuning-free" for the very important stochastic case.
> For all these reasons, we believe that our contributions (deriving a new stochastic adaptation of their method, show both theoretically and empirically that it fixes the issue above) are significant as we discuss in details in Section 1.1.

---

### Official Review · Reviewer_wGNE · 2025-11-04

**Soundness:** 3
**Presentation:** 3
**Contribution:** 2
**Rating:** 6
**Confidence:** 4

**Summary:**

The paper, proposed an efficient stochastic variant of the deterministic AdaGD of Malitsky and Mischchenko (2020). The proposed method does require any hyper-parameter tuning, and the papers provides two main theorems one on almost sure convergence (theorem 3.2) for three settings/cases, and a theorem in convergence in expectation for strongly convex component functions. Experiments also show the performance of the proposed method.

**Strengths:**

The paper is well written and the idea is easy to follow. The authors did a very nice job in the narrative of the paper. The explanation of their main idea, difference with previous approach, and the theoretical analysis is clearly presented.

The results as far as i know, are the first of their kind as an analysis for the stochastic version of AdaGD of Malitsky and Mischchenko (2020) was never successfully proposed.

**Weaknesses:**

1. On Assumption 1:
What exactly does Assumption 1 mean? This looks like an artificial condition to impose, just to make the proof work. Why is it not in the statements of the main theorems? How can one guarantee that this is satisfied? Can the authors provide more explanation on this, and some even constructed examples that this assumption is satisfied?

2. The authors claim the following in Figure 1: "Our methods (AdaSGD, the blue curves) always perform well with small values of $\lambda_0$ and thus do not require step-size tuning to compete with optimally-tuned SGD and ..." This statement, by simply observing Figure 1 is clearly not accurate. For example ,for $\lambda =1$ or $\lambda =10$ then the method does not converge. This makes the method precisely parameter-dependent. For claiming a tuning-free algorithm, one needs to have full independence in any parameter choice.

3. I would appreciate it if the labels were provided for any figure, not just Figure 2, which forces the reader to go back and forth between two figures to understand what is happening.

4. On presentation: the paper devotes 1.5 pages to related work (not closely related to the paper) and then two full pages to the lemmas related to bounding specific quantities. I find this unnecessary, as I would like to have this space for extra numerical evaluation of the method and convergence analysis. The authors decided to move to Appendix A, the important information on what classes of problems they really solve, which, in my opinion, should have been in the main paper (instead of the lemmas on bounding quantities that can be moved to the appendix)

**Questions:**

Please see Weaknesses

---

> ### Author Response · Authors · 2025-11-20
>
> ## On Assumption 1:
>
> Assumption 1 is stated so that the step taken by SGD remains an unbiased estimator of true gradient descent step, i.e.
> so that the conditional expectation of $\lambda_k \nabla f_{\xi_k} (x_k)$ equals $\lambda_k \nabla f(x_k)$
> (see also Equation 8).
> We would like to clarify that this Assumption is not artificial, as **the step-size** $\lambda_k$ **we propose is exactly made to satisfy this property**. This is why it is not assumed in the main theorems, because at this stage, it is proven true (hence no need to assume it anymore).
> It is only made in the beginning of the paper to get the most general derivations on possible choices of $\lambda_k$.
> We added a remark (Remark 2.1) in the paper to emphasize this.
>
> ## On the sensitivity to $\lambda_0$
> While we may agree with the reviewer that the method is not fully "parameter-free" in the sense that it may not work well for large $\lambda_0$, we strongly stand behind the fact that the method is **tuning free** in the sense that taking a small value of $\lambda_0$ is always at least as good as a large one. In contrast, Figures 1 and 2, highlight that for concurrent methods, the step-size must not only be too large, but also not too small. Indeed, the sensitivity curves for concurrent methods displayed in  Figures 1 and 2 have a "U-shape" which results in an optimal value of $\lambda_0$ that is problem-dependent for these methods, while taking small $\lambda_0$ always work for our AdaSGD method.
>
> In fact, the exact same argument is made by [Malitsky and Mishchenko, 2019] in the deterministic setting. We added a new comment (Remark 1.1) in the paper to clarify this point.
>
> ## Miscelleanous
> * We addded the labels to all figures as advised by the reviewer.
> * The cases studied in the paper are detailed in the Appendix but also discussed in the beginning of Section 3.1. We made the technical computations of Section~2 part of the main text because they constitute what makes the method novel and represent the key contribution to adapt [Malitsky and Mishchenko, 2019] to the stochastic setting.

---

### Author Response · Authors · 2025-11-20

# General comment to the AC

We would like to **warmly thank all reviewers** for their feedback. Due to the current situation we would now like to summarize what we have done to answer their concerns.



The reviewers' main requests revolved around additional empirical evidence of the performance of our method and clarification on the importance of our contribution. We now addressed both these points (below and in the new version). Although there is no opportunity for further discussion, we believed the clarification and additional content answered all their queries.


We provided **detailed answers to all the concerns** and questions raised by the reviewers in dedicated answers to each of them.
We also have uploaded **a new PDF version of the paper**. We highligted these changes in blue font in the new version.


In short:
* Several reviewers' acknowledged the novelty, and theoretical soundness of our work, and that adapting the AdaGD algorithm (Malitsky and Mischchenko, 2020) to the stochastic setting was a notable achievement.
* Two of them expressed a need to further explain the importance and signifiance of our contribution. We adressed this by clarifying how important it is to provide adaptive variants of SGD that do not require **any** hyper-parameter tuning, nor further knowledge on the objective function. This is of high interest to the community given the prominence of SGD for training machine learning models.
* For the signifiance of the contribution we emphasize that the prior work of Malitsky and Mischchenko (2020) received a significant echo in the deterministic setting. Depsite this, it had not been as successfully adapted to the stochastic setting. This is our main contribution, and we have better highlited it in our revised version.
* The reviewers asked for further experiments, moslty revolving around error-bars and further comparison with other methods. We added the additional experiments suggested by the reviewers to the paper. That is in particular: multiple runs with error bars for each figure, comparison to additional adaptive baselines (Polyak step-size and step-size Adagrad), effect of different batch-sizes.

---

### Meta-Review · Area_Chair_oVgC · 2026-01-07

**Summary:**

The paper presents a stochastic variant of the deterministic AdaGD (Malitsky and Mischchenko, 2020). The reviewers appreciate the theoretical contribution of the paper, while they list some remaining weaknesses.

**Reviewer Concerns:**

One reviewer noted that the presentation of the paper can be improved. Examples include explanation for Assumption 1, the related work section, and Appendix A. In addition, the explanation and justification for Figure 1 are not very clear. The authors should explain more about the theoretical settings of the paper and the scope to which their method applies.

The experiments are limited, and the reviewers suggest repetitions of stochastic experiments to ensure consistent results.  Also, the additional computational cost should be discussed and compared. Two reviewers share concerns about the significance/originality of the work.

In summary, while the reviewers noted some of the paper's contributions, there are things to improve for this paper.

**Reviewer Scores:**

The reviewers did not reply in the initial stage, and there are no major misunderstanding of the paper. Thus, I do not think the reviewers' scores would change too much after the rebuttal.

---

### Decision · Program_Chairs · 2026-01-26

Reject